

# Climatology of the Mount Brown South ice core site in East Antarctica: implications for the interpretation of a water isotope record

Sarah L. Jackson[1,2,3], Tessa R. Vance[4], Camilla Crockart[4], Andrey Moy[5,4], Christopher Plummer[4], Nerilie J. Abram[1,2,3]

[1]Research. School of Earth Sciences, Australian National University, Canberra ACT 2601, Australia
[2]Australian Centre for Excellence in Antarctic Science, Australian National University, Canberra ACT 2601, Australia
[3]ARC Centre of Excellence for Climate Extremes, Australian National University, Canberra ACT 2601, Australia
[4]Australian Antarctic Program Partnership, Institute for Marine & Antarctic Studies, University of Tasmania, Hobart TAS 7004, Australia
[5]Australian Antarctic Division, Department of Climate Change, Energy, the Environment and Water, Kingston TAS 7050, Australia

*Correspondence to*: Sarah L. Jackson (sarah.jackson@anu.edu.au)

**Abstract.** Water stable isotope records from ice cores ($\delta^{18}$O and $\delta$D) are a critical tool for constraining long-term temperature variability in the high-latitudes. However, precipitation in Antarctica consists of semi-continuous small events and intermittent extreme events. In regions of high-accumulation, this can bias ice core records towards recording the synoptic climate conditions present during extreme precipitation events. In this study we utilise a combination of ice core data, re-analysis products and models to understand how precipitation intermittency impacts the temperature records preserved in an ice core from Mount Brown South in East Antarctica. Extreme precipitation events represent only the largest 10% of all precipitation events, but they account for 44% of the total annual snowfall at this site leading to an over-representation of these events in the ice core record. Extreme precipitation events are associated with high-pressure systems in the mid-latitudes which cause increased transport of warm and moist air from the southern Indian Ocean to the ice core site. Warm temperatures associated with these events result in a +2.8° C warm bias in the mean annual temperature when weighted by daily precipitation, and water isotopes in the Mount Brown South ice core are shown to be significantly correlated with local temperature when this precipitation-induced temperature bias is included. The Mount Brown South water isotope record spans more than 1000 years and will provide a valuable regional reconstruction of long-term temperature and hydroclimate variability in the data-sparse southern Indian Ocean region.



## 1 Introduction

Antarctica and the southern high-latitudes play a critical role in the global climate system through polar-tropical teleconnections (Li et al., 2021; Yuan et al., 2018) and the modulation of meridional transport of heat (Trenberth and Caron, 2001; Turner et al., 2014). Despite the global importance of Antarctica to the climate system, direct observations of Antarctic climate are largely restricted to the satellite-era (1979-onwards; Jones and Lister, 2015), and limited longer-term data from permanent land-based
stations (Turner et al., 2005, 2020). The short observational window limits our ability to understand the natural variability of Antarctic climate at decadal- to millennial-scales. As such, it is necessary to employ paleo-climate archives to extend observational records and better constrain natural variability. Ice cores have been utilised across the continent to provide both low-resolution records of glacial-interglacial cycles and high-resolution records of inter-annual climate variability (e.g. Augustin *et al.*,
2004; Stenni *et al.*, 2017). Coastal or near-coastal ice cores located in regions of high-accumulation are increasingly being used to help frame modern anthropogenic warming in the context of natural climate variability in Antarctica, as these can provide annually-resolved records capable of recording inter-annual to decadal fluctuations in the climate.

Water stable isotope ratios in ice core records ($\delta^{18}O$ and $\delta D$) are routinely used to reconstruct past temperature variations across Antarctica (e.g. Dansgaard, 1964; Stenni *et al.*, 2017). Fractionation of water isotopes occurs as heavy isotopes are preferentially removed as air masses are transported poleward, leading to a strong spatial isotope-temperature relationship across Antarctica (Lorius et al., 1969; Masson-Delmotte et al., 2008). However, water isotope records are controlled not only by site-
temperature. Source region conditions (Markle and Steig, 2022), transportation pathways (Markle et al., 2012; Jouzel et al., 1997) and post-depositional effects (Casado et al., 2018) also influence the water isotope record, leading to complications in interpreting the climate signature captured by stable water isotopes in ice cores. Similarly, water isotopes primarily reflect conditions during precipitation events, and can be strongly affected by the intermittency of precipitation. This can result in potential signal bias
where climate during certain synoptic conditions is preserved and recorded in the accumulated snowfall which may not necessarily be representative of the mean climatology (Turner et al., 2019).

Snowfall across Antarctica manifests as a combination of frequent small events and infrequent large events. We refer to the infrequent large events as extreme precipitation events, or EPEs and define these
as representing days where daily snowfall amount is in the 90th percentile or higher. Although infrequent, EPEs can contribute a significant amount of the annual snowfall at a particular site, and are the primary drivers of seasonal and inter-annual accumulation variability in many regions of Antarctica (Turner et al., 2019). The water isotope record from ice cores is primarily an archive of climate conditions present during precipitation, thus it is critical to understand the biases introduced to the
record due to variability in EPEs and precipitation intermittency. At a continental scale, more than 40% of total precipitation can be attributed to EPEs (Turner et al., 2019). In many coastal regions, where



orography exerts stronger controls on the transport of marine air masses inland, EPEs can have an even larger impact. For example, in the Amery Ice Shelf region on average 60% of annual snowfall accumulates during only 10 days each year (Turner et al., 2019). Similarly, at Aurora Basin North, an
ice core site located on the East Antarctic Plateau approximately 500 km inland from the coastal station Casey, it is estimated that half of the annual precipitation occurs during only 10% of days (Servettaz et al., 2020).

Atmospheric rivers are a subset of EPEs and can greatly influence the total annual accumulation. In
Antarctica, these occur when there is a narrow band of strong horizontal water vapour transport which brings warm, moist air from the low- to mid-latitudes of the southern hemisphere to the cold Antarctic region. A study by Wille et al., (2021) identified that 35-45% of all EPEs (when applying a definition of the 95[th] percentile) in Antarctica can typically be classified as atmospheric rivers. This percentage increases when more stringent classifications are imposed on EPEs. Similarly, Adusumilli et al. (2021)
found that 37-55% of all EPEs in 2019 (again defined as 95[th] percentile) on the West Antarctic Ice Sheet could be attributed to atmospheric rivers. In 2009 and 2011, Princess Elizabeth Station in Dronning Maud Land received 74% and 80% of total annual snowfall respectively from a series of atmospheric river events, resulting in anomalously high snowfall in these years (Gorodetskaya et al., 2014).

Annually-resolved ice core records are a critical tool for improving our understanding of natural variability in the Earth's climate (Stenni et al., 2017). However, there are currently only a few published records from long (multi-centennial), high-resolution ice cores in the Indian Ocean sector of Antarctica, which covers a broad region between Enderby Land and Wilkes Land in East Antarctica (67° E-160° E;
Delmotte et al., 2000; Ekaykin et al., 2017; Stenni et al., 2017). Previous work on the Law Dome ice core trace chemistry record has indicated that there are strong teleconnections between the southern Indian Ocean climate and Australian hydroclimate (Udy et al., 2021). The Law Dome snow accumulation record (Roberts et al., 2015) has been used to reconstruct rainfall across southwest Western Australia, indicating that current drying in the region is unusual but not unprecedented over the
past 2000 years (Zheng et al., 2021; van Ommen and Morgan, 2010). Similarly, there is a strong correlation between concentrations of aerosol sea salts during summer snowfall at Law Dome with rainfall in eastern Australia (Vance et al., 2013, 2015; Udy et al., 2022). This relationship has been used to highlight the increased drought risks in eastern Australia associated with positive phases of the Interdecadal Pacific Oscillation and the potential for megadroughts much longer than any historically
observed drought in this region (Vance et al., 2015, 2022).

An extensive site-selection study by Vance et al., (2016) identified Mount Brown South as a promising location for a new ice core that would likely provide unique climate signals and be complementary to the Law Dome ice core record. The authors also suggest that an ice core collected from Mount Brown
South would contain strong teleconnections to the mid-latitudes of the southern Indian Ocean. In 2017/2018, a series of four ice cores were drilled at this site, including a 295m long main ice core record, which is estimated based on a current age scale in development to extend back 1200 years. Early investigations into the record by Crockart et al., (2021) demonstrated that the ice core preserves an



annually resolved climate history that differs from the Law Dome record, with a clear signal of mid-
latitude Indian Ocean atmospheric variability in annual sea salt concentrations relating to the El Niño-
Southern Oscillation.

In this study, we investigate how the synoptic climate conditions associated with extreme precipitation
at the Mount Brown South site impact the ice core record. Firstly, we investigated seasonal and inter-
annual variability in precipitation and extreme precipitation events using the Regional Atmospheric
Climate Model, RACMO2.3p2. Secondly, we identified geopotential height and temperature anomalies
associated with extreme events using ERA-5. Thirdly, we discuss how temperature anomalies
associated with extreme precipitation lead to a temperature bias in the Mount Brown South water
isotope.

**2 Methods**

**2.1 Mount Brown South ice core site properties, analysis, and timescale**
**2.1.1 Site description**

During the 2017/2018 Australian Antarctic Program (AAP) field season, four ice cores were drilled at
69.11° S 86.31° E, south of Mount Brown in Wilhelm II land in East Antarctica, at an altitude of 2,084
m a.s.l. (Fig. 1). The drill-site (hereafter referred to as Mount Brown South or MBS) is located
approximately 380 km east from Australia's Davis Station and approximately 1,000 km west of Law
Dome. The four cores included a 295 m long core (hereafter referred to as MBS-Main) and three short
surface cores (MBS-Alpha, MBS-Bravo and MBS-Charlie) which were each 20-25 m in length. MBS-
Main was drilled from 4 m below the snow surface using a Hans Tausen drill (Johnsen et al., 2007;
Sheldon et al., 2014), and MBS-Alpha, MBS-Bravo and MBS-Charlie were all drilled from the surface
using a Kovacs drill. The MBS site is located approximately 15 km from where a series of firn cores
were drilled between 1997-1999 (Foster et al., 2006; Smith et al., 2002).

The four cores were cut into 1 m long sections on site, and stored in individual LDPE Poly bags. The
cores were transported by helicopter from the drill site to freezer storage at Davis Station and then
transported by ship to Hobart. The ice cores were stored in the Australian Antarctic Division
commercial cold storage facility and further processing and analysis was completed at the Institute for
Marine and Antarctic Studies (IMAS) in Hobart.



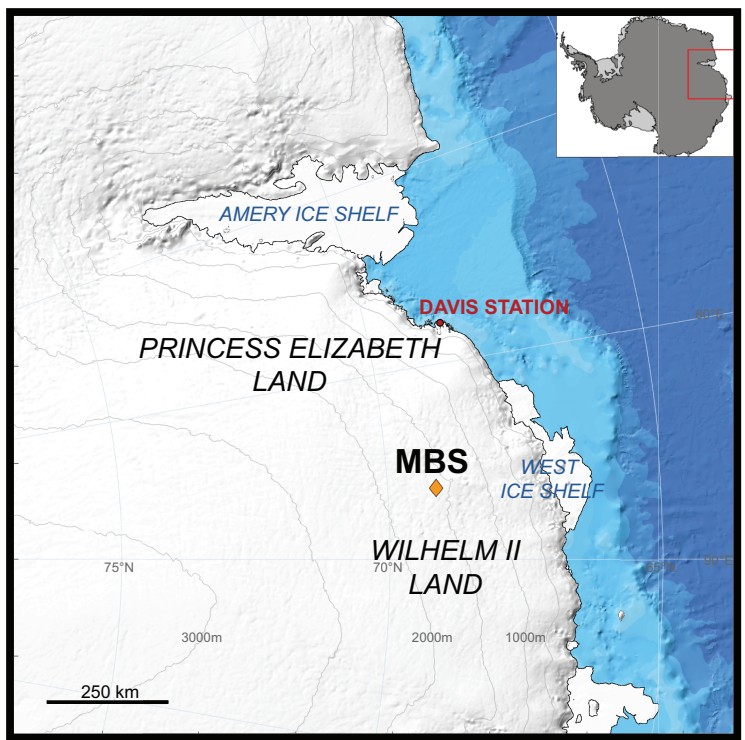

**Figure 1: The Mount Brown South (MBS) ice core drill site, located in Wilhelm II Land in East Antarctica. The location of the drill site is indicated by the yellow diamond. The map was produced using Antarctic Mapping Tools (Greene et al., 2017) with ice sheet elevation data from Bedmap2 (Fretwell et al., 2013) and bathymetric data from IBSCO (Arndt et al., 2013).**

### 2.1.2 Ice core analyses

The MBS-Main and MBS-Charlie ice cores were cut into discrete samples for water isotope analysis. Analyses for MBS-Main are complete and a full age scale is being prepared for publication. Only the upper 4–20 m of MBS-Main and 0–20 m of MBS-Charlie will be discussed in this paper, as these sections overlap with the modern satellite era (1979-present) and have been analysed for water isotopes. For MBS-Main, 3 cm samples were cut and analysed for the 4 – 16 m section, and 1.5 cm samples were analysed for 16–20 m section. MBS-Charlie was cut uniformly into 3 cm samples. The samples were initially analysed for deuterium isotope ($\delta D$) and oxygen isotope ($\delta^{18}O$) ratios by cavity ring-down spectroscopy using a Picarro L2130-$i$ water isotope analyser in Hobart. The secondary parameter $d_{excess}$ was also calculated based on $\delta D$ and $\delta^{18}O$ measurements. The analysed samples were re-frozen and stored at -18° C and later transported frozen to the Australian National University (ANU) in Canberra. At the ANU, samples were re-melted and re-analysed on a Picarro L2140-$i$ to include $\delta^{17}O$ and the secondary parameters $^{17}O_{excess}$ and $d_{ln}$. There is excellent inter-laboratory agreement between $\delta D$, $\delta^{18}O$ and $d_{excess}$ from the initial measurements in Hobart and the repeat measurements at the ANU which gives confidence that fractionation was negligible during the melting and re-freezing processes in the laboratories ($r > 0.99$ for all parameters).





Discrete chemistry analyses of MBS-Main core, MBS-Charlie and MBS-Alpha were also conducted at IMAS on 3 cm resolution samples. The MBS-Bravo core was collected exclusively for persistent
organic pollutant analyses that are still ongoing and therefore will not be discussed as part of this study. Analyses of anions and cations were performed using a Thermo-Fisher/Dionex ICS3000 ion chromatograph. A detailed discussion of the discrete chemistry analyses can be found in Crockart et al., (2021).

### 2.1.3 Ice core dating

Chemical species with clear seasonality (i.e. non sea salt sulphate (nssSO$_4^{2-}$), sodium (Na$^+$), the ratio of sulphate to chloride (SO$_4^{2-}$/Cl$^-$)) were used in conjunction with water isotope measurements to identify annual layers in MBS-Alpha, MBS-Charlie and the upper portion of MBS-Main. The Pinatubo eruption
(mid-1991) was identified as a peak in nssSO$_4^{2-}$ in all cores and was used as a marker to confirm the accuracy of layer counting. Layer thicknesses were combined with an empirical density model to determine annual ice equivalent snow accumulation rates. Further details of the dating procedures and accumulation calculations for the short cores and upper section of MBS-Main can be found in Crockart et al., 2021.

### 2.2 Re-analysis and model data

### 2.2.1 RACMO2.3p2

We use total daily precipitation output from the Regional Atmospheric Climate Model, RACMO2.3p2 (Wessem et al., 2014) to examine trends in precipitation at the MBS site for the period 1979-2016 RACMO is a high-resolution atmosphere-only regional model with a horizontal resolution of approximately 27 km. It has been demonstrated to accurately reproduce observational surface mass balance (SMB) changes across Antarctica (Lenaerts et al., 2013; Marshall et al., 2017). RACMO2.3p2
is forced by ERA-Interim every 6 hours at the lateral and upper-atmospheric boundaries. To maintain consistency with previous studies on Extreme Precipitation Events (Turner et al., 2019), a precipitation day was identified when total daily precipitation exceeded 0.02 mm WE (water equivalent). Extreme Precipitation Events (EPEs) were classified as those exceeding the 90$^{th}$ percentile of all precipitation days at MBS, as defined by Turner et al., 2019. According to this definition, an EPE at MBS is any day
where daily precipitation values exceed 2.7 mm day$^{-1}$ WE.

RACMO2.3p2 has been demonstrated to reproduce seasonal and inter-annual variability in precipitation in Antarctica with a high degree of accuracy (Wang et al., 2016; Mottram et al., 2021). We compared monthly precipitation output from RACMO2.3p2 with monthly precipitation from ERA-5 and surface
mass balance estimates from Modèle Atmosphérique Régional (MARv3.9). We found that all three datasets were highly-correlated (Table A1 in the Appendix), although surface mass balance estimates from MARv3.9 are lower for the summer months of December and January likely due to included estimations of mass loss in the surface mass balance calculation (Fig. A1 in the Appendix). We chose



here to use RACMO2.3p2 as it has been demonstrated to accurately reproduce precipitation across
Antarctica and to allow for direct comparisons with previous work on EPEs (Turner et al., 2019)

**2.2.2 ERA-5**

Output from the reanalysis product ERA-5 (produced by the European Centre for Medium-Range
Weather Forecasts) was used to investigate the climatic conditions associated with extreme precipitation
events at MBS. ERA-5 provides hourly data at 0.25° x 0.25° resolution for a number of atmospheric,
sea and land-based processes from 1950 onwards. The quality of the data is improved from 1979 as
satellite-based measurements help to reduce uncertainties in the observational data. This is particularly
true for the data-sparse southern high-latitudes, hence we only consider the period from 1979-2016 in
our analyses.

Daily 500 hPa geopotential height was extracted from ERA-5 for the entire region south of 40° S.
Geopotential height anomalies were calculated for each day relative to the seasonal average for that day.
We calculate the seasonal average as the mean value for each day from 1979-2016, with a 30-day
rolling mean filter applied across daily-mean values to obtain a smoothed seasonal record (Servettaz et
al., 2020).

Daily surface temperature and 650 hPa temperatures were also extracted from ERA-5 for the entire
region south of 40° S. In Antarctica, an inversion layer results in surface temperatures that are
frequently cooler than temperatures above the inversion layer (Jouzel and Merlivat, 1984), where
condensation primarily occurs. Variability in water isotopes in ice cores reflect changes in condensation
temperatures rather than changes in surface temperature, and so we extracted temperatures at the 650
hPa pressure level from ERA-5 to approximate condensation temperatures at MBS. The mean annual
surface pressure measured at a nearby Automated Weather Station (AWS) is 750 hPa, which is located
19 km away from the ice core site. Thus we selected the 650 hPa pressure level as the closest
representative to the level where atmospheric pressure is 90% of surface pressure for the MBS site, and
is considered to be permanently above the inversion layer (Servettaz et al., 2020). Temperature
anomalies were calculated by the same method described above for 500 hPa geopotential height
anomalies.

Although RACMO2.3p2 is forced by data from ERA-Interim, we use geopotential height and
temperature data from ERA-5, as this product replaces ERA-Interim with both higher spatial and
temporal resolution. ERA-5 has been demonstrated to reproduce observational data well over many
regions of Antarctica (Tetzner et al., 2019; Zhu et al., 2021). To confirm the validity of ERA-5
temperature data for this location, we compared ERA-5 temperature measurements to temperature
measurements from the nearby AWS discussed above (located at 69.13° S, 86.00° E at an elevation of
2078 m). Hourly temperature data is available from the AWS from 2003-2008. Comparisons of monthly
mean temperature from ERA-5 for the nearest grid-cell to the AWS indicate that ERA-5 is able to
reproduce the AWS temperature observations well (r = 0.99). There is a warm bias in the ERA-5 data



(mean difference = 1.4° C), (Fig. A2 in the Appendix), which may be due to localised temperature effects at the AWS that are not resolved by ERA-5.

## 2.3 HySPLIT back-trajectory modelling

### 2.3.1 Model description


The Hybrid Single-Particle Lagrangian Integrated Trajectory model (HYSPLIT) was developed by the National Oceanic and Atmospheric Administration's (NOAA) Air Research Laboratory as a tool for constructing 3-D air parcel trajectories (Stein et al., 2015). Here, we use HySPLIT to generate 5-day

back-trajectories (120 hours), originating at the MBS site at a height of 1500 m above ground level, which is equivalent to approximately 3500 m above sea level. HySPLIT was developed to be forced using meteorological conditions defined by the NCEP/NCAR re-analysis product, although boundary conditions may also be defined by different re-analysis products, with ERA-Interim frequently used. However, the software is formatted to readily utilise NCEP/NCAR datasets, while other re-analysis

products require re-formatting. Sinclair et al., (2013) compared the difference in back-trajectories for an ice core site in the Ross Sea region of Antarctica forced by both ERA-Interim and NCEP/NCAR and found the results comparable. As such, for this study the meteorological parameters in the HySPLIT model were forced using the more readily applied NCEP/NCAR Global Reanalysis data. Daily trajectories were generated from 1 January 1979 to 31 December 2019, resulting in a total of 14,610

back-trajectories.

### 2.3.2 HySPLIT clustering

Individual trajectories were clustered using HYSPLIT's inbuilt clustering algorithm, which aims to

minimise inter-cluster variability while maximising intra-cluster variability for a user-defined number of clusters (Stein et al., 2015). Due to computational limitations, the full dataset was clustered at 2-day resolution (7160 trajectories) using points at 4-hour intervals along the trajectory. The HySPLIT clustering algorithm requires the user to define the number of clusters, which is chosen to minimise the total spatial variance while still capturing a variety of different trajectories linked to synoptic climate

conditions. We identified a total of 5 clusters; this number was chosen as it was the point where total spatial variance approached a minimum, but it still allowed for the identification of distinct trajectories linked to synoptic climate conditions.

## 3. Results and discussion


### 3.1. Seasonal and interannual accumulation

The mean annual accumulation from 1979-2016 at MBS from RACMO2.3p2 is 0.238 ± 0.035 m WE (water equivalent) year$^{-1}$, with uncertainties representing 1 standard deviation (Fig. 2). The mean

measured annual accumulation rates for MBS-Alpha, MBS-Charlie and MBS-Main are 0.273 ± 0.06, 0.271 ± 0.07 and 0.284 ± 0.07 m WE yr$^{-1}$ respectively. Accumulation rates have been converted to water



equivalent accumulation from the original Crockart et al. (2021) data which is presented as ice equivalent, assuming an ice density of $\rho = 917$ kg m$^{-3}$. Mean annual accumulation from RACMO2.3p2 slightly underestimates accumulation rates derived from the ice cores, however the annual accumulation
from the model is still significantly correlated with ice core accumulation (Table 1), with the exception of MBS-Charlie which has been previously shown to be poorly correlated with accumulation (Crockart et al., 2021). As noted in Crockart et al., (2021), there is potential for annual layers to be missed during dating, particularly during low-accumulation years. For example, model outputs identify 1993/1994 as sequential years with low accumulation (from RACMO2.3p2 in this study, and in ERA-5 and MAR in
Crockart et al., 2021) while accumulation measurements from MBS-Main, MBS-Alpha and MBS-Charlie do not identify consecutive low-accumulation years during this period (Fig. 2). This introduces a degree of dating uncertainty into our accumulation measurements which may be responsible for the poor correlation between MBS-Charlie and RACMO2.3p2 accumulation. Despite these uncertainties, MBS-Alpha, MBS-Main and the site-averaged accumulation record are all significantly correlated with
RACMO2.3p2 accumulation and we therefore consider the model to reliably capture the interannual variability of accumulation at the MBS site.

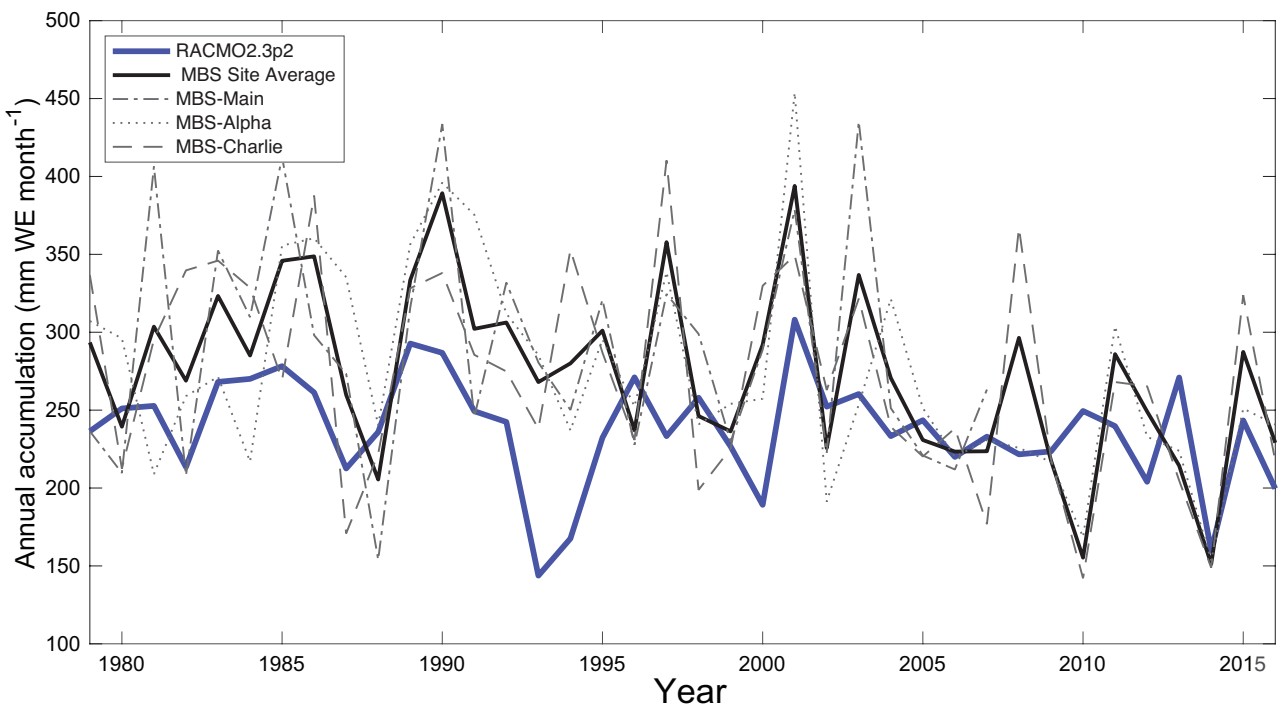

**Figure 2: A comparison of annual accumulation from RACMO2.3p2 (solid blue line) and the MBS ice core records. Site averaged**
**accumulation is shown by the solid black line, with the individual records indicated by dashed lines.**



| | Pearson's Correlation Coefficient | p-value | Time-period |
|---|---|---|---|
| MBS-Main | **0.433** | **0.019** | 1979-2007 |
| MBS-Alpha | **0.436** | **0.006** | 1979-2016 |
| MBS-Charlie | 0.230 | 0.165 | 1979-2016 |
| MBS-Site | **0.453** | **0.004** | 1979-2016 |

**Table 1: Pearson's correlation coefficient (r) and p-values for mean annual accumulation from RACMO2.3p2 with the MBS-Main, MBS-Alpha, and MBS-Charlie ice cores and the ice core site average. Significant correlations (p < 0.05) are marked in bold and are found for all ice cores aside from MBS-Charlie.**

EPEs strongly impact the annual accumulation at MBS (Fig. 3a). There are only $23.2 \pm 6.2$ EPE events per year (i.e. days where total daily accumulation exceeds 2.7 mm WE day$^{-1}$) from 1979-2016 ($6.3 \pm 1.7$ % of days). However, these small number of days account for $43.5 \pm 8.3$ % of the total annual snowfall.
This implies that the synoptic conditions associated with extreme moisture transport to the ice core site will be overrepresented in annual snowfall, and therefore also overrepresented in the ice core record.

The highest accumulation from RACMO2.3p2 occurs during the winter months at MBS (May = 24.1 mm WE month$^{-1}$, June = 23.0 mm WE month$^{-1}$, July = 23.5 mm WE month$^{-1}$), with a secondary peak in
November (21.3 mm WE month$^{-1}$) giving a bi-modal character to the annual cycle of accumulation (Fig. 3c). Accumulation from non-EPE snowfall is relatively constant throughout the year ($10.9 \pm 1.3$ mm WE month$^{-1}$), with the seasonality largely driven by EPEs. Bimodality is observed in both the mean number of EPE days in a month and the amount of accumulation associated with EPEs. May, June and July have 2.6, 2.7 and 2.4 EPEs per month respectively, while November has 2.6. This results in
maxima in accumulation from EPEs of 11.0 mm WE month$^{-1}$ for May, 11.2 mm WE month$^{-1}$ for June, 10.8 mm WE month$^{-1}$ for July, and 11.8 mm WE month$^{-1}$ for November. Minimum total monthly accumulation (14.0 mm WE month$^{-1}$), accumulation from EPEs (4.5 mm IE month$^{-1}$) and number of EPE days (0.9) occurs in January. 89.5% of the variance in monthly accumulation can be explained by variance in accumulation from EPEs ($r^2 = 0.895$).

Variability in accumulation at an inter-annual scale is also driven by variability in EPEs (Fig. 3b). The contribution of smaller (i.e. not EPE) precipitation events to annual accumulation is relatively constant from 1979-2016 ($0.131 \pm 0.016$ m WE yr$^{-1}$). In contrast, there is considerable interannual variability in both the number of EPE events in a year ($23.2 \pm 6.2$ days yr$^{-1}$) and the amount of accumulation from
EPEs in a year ($0.105 \pm 0.031$ m WE yr$^{-1}$). The number of EPE days in a year is strongly correlated with both total annual accumulation (r = 0.90, p = <0.001) and accumulation from EPEs (r = 0.97, p = <0.001). 85% of the variance of interannual accumulation can be explained by variance in accumulation from EPEs ($r^2 = 0.85$).





**Figure 3: a) Annual accumulation at MBS from RACMO2.3p2 for 1979-2016. Accumulation from non-EPE precipitation days is shown in dark blue, and accumulation from EPE's in light blue. The blue diamonds indicate the number of EPE days in a year. b) Annual accumulation at MBS versus number of extreme precipitation days in a year. The dashed line indicates the linear regression (r = 0.90). c) Mean monthly precipitation at MBS from RACMO2.3p2 for 1979-2016. Colours are as in a). Blue diamonds represent the mean number of EPE days per month, with error bars indicating the upper and lower quartiles.**

## 3.2 Climatic conditions during EPEs

### 3.2.1 Geopotential height anomalies

It has previously been noted that extreme precipitation across Antarctica is frequently associated with geopotential height anomalies that cause atmospheric blocking, which forces warm moist air from the mid-latitudes onto the Antarctic continent (Turner et al., 2019; Servettaz et al., 2020). The 500 hPa geopotential height field was investigated using ERA-5 (Section 2.2.2) to understand the synoptic climate conditions associated with extreme precipitation at MBS, and to determine whether similar blocking events could be identified during EPEs at this location.





We calculated the mean 500 hPa geopotential height anomaly for all days that were identified as an extreme precipitation event during 1979-2016 (Section 2.4.2). We found that extreme precipitation was on average associated with a strong positive geopotential height anomaly located at 100-120° E, 55-65° S, offshore and to the east of the ice core site. An associated weak negative pressure centre can also be identified to the west of the ice core site (Fig. 4a).

The location of high-pressure system during EPEs, directly to the east of the MBS ice core site, acts to block the transport of air masses from the west. Instead, the high-pressure system directs meridional transport of warm, moist air up onto the continent, where rapid cooling can induce large precipitation events. Similar studies at other locations in East Antarctica have also demonstrated that high-pressure blocking systems located to the east of ice core sites are related to high levels of precipitation at the respective ice core sites (Servettaz et al., 2020; Turner et al., 2019; Scarchilli et al., 2011).

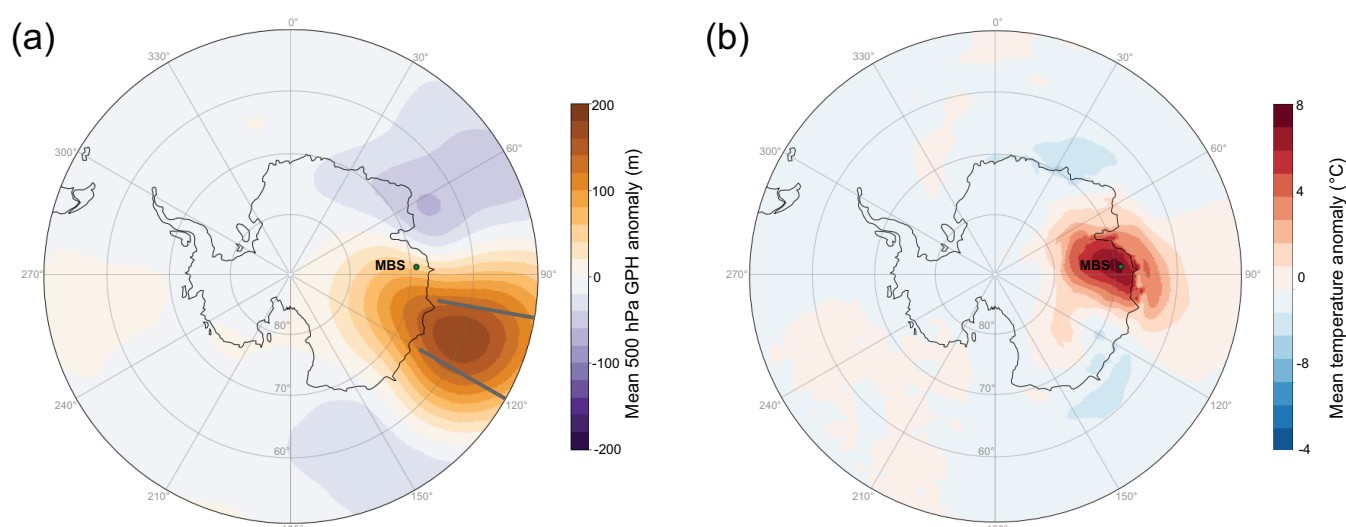

**Figure 4: a) Mean 500 hPa geopotential height anomaly relative to the seasonal mean from ERA-5 for all EPE days identified using RACMO2.3p2 from 1979-2016. Grey bars indicate the regions between which the blocking index is calculated. b) Mean surface temperature anomaly relative to the seasonal mean from ERA-5 for all EPE days identified using RACMO2.3p2 from 1979-2016.**

### 3.2.2 Comparison with synoptic studies

Previous studies have categorised synoptic-scale climate conditions in the southern Indian Ocean using self-organising maps (Udy et al., 2021) and *k*-means clustering methods (Pohl et al., 2021). These methods utilise 500 hPa geopotential height anomalies (in Udy et al., 2021) and 700 hPa geopotential height anomalies (in Pohl et al., 2021) to identify regional-scale synoptic patterns. Synoptic type 1 (SOM1) from Udy et al. (2021) displays a geopotential height anomaly remarkably similar to that





observed during extreme precipitation events at MBS – that is, a strong positive geopotential height
anomaly located to the east of the ice core site (at approximately 100-120° E), coupled with a low-
pressure centre to the west. We filtered the self-organising maps from Udy et al., 2021 to examine the
proportion of each synoptic type associated with EPEs at MBS and found that there is a strong
preference for extreme snowfall to occur during SOM1 (34.4 % of EPEs occur during SOM1 - Table
A2 in the Appendix). When more stringent filters are used to define extreme precipitation events (i.e.
when EPEs are defined as the 95[th] and 99[th] percentiles of precipitation) there is a further preference for
these events to be associated with SOM1 (41.0 % and 53.3 % respectively). This is consistent with the
results discussed in Section 3.1.1, with both approaches indicating that the presence of a high-pressure
system to the east of the ice core site is the primary driver of extreme precipitation at MBS.

**3.2.3 Blocking index**

To investigate the impacts of atmospheric blocking on accumulation and EPEs at MBS, we compared
total seasonal accumulation, EPE-associated seasonal accumulation, and non-EPE seasonal
accumulation with a blocking index initially derived by Wright (1974) and later modified by Pook and
Gibson (1999; Table 2). This blocking index uses the NCEP-NCAR dataset to calculate the difference
in the sum of geostrophic westerly winds at relatively low latitudes (25-30° S) and high latitudes (55-
60° S) with the sum of mid-latitude westerly winds (40-50° S), and can be defined as:

$$BI = 0.5(U_{25} + U_{30} + U_{55} + U_{60} - U_{40} - U_{50} - 2U_{45})$$

where $U_x$ represents the zonal component of mean 500 hPa wind at latitude x. A high BI value indicates
either a reduction of the mid-latitude zonal winds or an increase in the high- and low-latitude zonal
winds (or a combination of both) and is indicative of atmospheric blocking.


| | SEASON | | | | | | | |
|---|---|---|---|---|---|---|---|---|
| | r | p | r | p | r | p | r | p |
| Total Accumulation | **0.362** | **0.02** | 0.132 | 0.42 | **0.338** | **0.04** | 0.261 | 0.11 |
| EPE Accumulation | **0.333** | **0.04** | 0.214 | 0.19 | 0.294 | 0.07 | **0.314** | **0.05** |
| Non-EPE Accumulation | 0.274 | 0.09 | -0.156 | 0.34 | 0.188 | 0.23 | -0.005 | 0.98 |

**Table 2: Pearson's correlation coefficient (r) and p-values for the zonally averaged blocking index (100–120° E) with total accumulation, EPE accumulation, and non-EPE accumulation for Dec–Jan (DJF), Mar–Apr (MAM), Jun–Aug (JJA), and Sep–Nov (SON). Significant correlations (p < 0.05) are shown in bold.**

Blocking indices are calculated based on monthly mean values and are averaged to generate seasonal
means. Here we calculate the blocking index between longitudes 100-120° E, as this is the region where





we identify a positive geopotential height anomaly associated with EPEs at MBS. Seasons are identified as DJF (austral summer), MAM (austral autumn), JJA (austral winter) and SON (austral spring).

For all seasons, there is no correlation between smaller (non-EPE) accumulation and blocking. This indicates that small-scale precipitation events at MBS can occur regardless of atmospheric blocking and are likely due to a variety of transport mechanisms. There is a strong positive correlation between EPE accumulation and atmospheric blocking in both SON and DJF indicating that large precipitative events during these seasons are driven by atmospheric blocking in the mid-latitudes. No correlation between
EPE accumulation is observed during MAM and JJA, but a positive correlation is noted between total accumulation and blocking in JJA.

This suggests that atmospheric blocking is an important mechanism for driving summer (DJF) and spring (SON) precipitation at MBS, as extreme precipitation is more likely to occur during these months
when there is mid-latitudinal blocking. We observe a slight poleward shift of the centre of the geopotential height anomaly associated with EPEs during the winter months (JJA; Fig. A3 in the Appendix). This suggests that atmospheric blocking may still be an important mechanism for precipitation during these months, but the poleward shift of the geopotential height anomaly may result in blocking at higher latitudes that are not captured by the Pook and Gibson (1999) blocking index.

**3.3 HySPLIT cluster analyses**

To further investigate the circulation patterns and synoptic conditions associated with EPEs, HySPLIT cluster analyses were used to identify how trajectory pathways are associated with EPEs.

**3.3.1 Cluster descriptions**

Using HySPLIT's inbuilt clustering algorithm, we identified 5 clusters which represent broad synoptic conditions for atmospheric transport to the MBS site (Section 2.3.2). The five clusters identified are as
follows: Cluster 1 is representative of locally sourced trajectories, with slow speed, a short path-length, and a moderate altitude along the full path-length (36.7 % of all trajectories; Fig. 5a). Cluster 2 (26.5 % of all trajectories) represents a high-altitude easterly pathway that follows the coastline of Wilkes Land. Cluster 3 (22.8 % of all trajectories) follows a high-altitude westerly pathway from the western Southern Indian Ocean and coastal Queen Maud Land. Cluster 4 (9.5 % of all trajectories) represents a
cyclonic pathway that originates in the Southern Indian Ocean. It remains at a low altitude (< 1000 m AGL) for most of the pathway, before increasing in altitude as it moves up onto the Antarctic continent (Fig. 5b). The remaining 4.4% of all trajectories are part of the trans-continental pathway defined by cluster 5.

Clusters 1, 4, and 5 only have weak seasonality (i.e. a similar proportion of trajectories is associated with each cluster for each season). Clusters 2 and 3 show stronger seasonality. Few trajectories are associated with cluster 3 during the austral summer (2.6% in DJF) compared with autumn (8.1% in MAM). In contrast, there are a greater proportion of trajectories associated with cluster 2 during the





austral summer (10.0 %) than the other seasons (4.4 – 6.3 %). It is important to note that the clusters
represent a continuous spectrum of trajectories moving towards the MBS site rather than discrete
pathways, but they still provide a useful tool for understanding the variability in moisture transport
pathways.

**Figure 5: a) The 5 different clusters identified by HySPLITs back-trajectory clustering algorithm for all back-trajectories**
**originating at the MBS ice core site from 1979-2016. The MBS ice core site is indicated by a yellow star. b) Mean altitude of each**
**cluster for the full 120 hours back-trajectory, with time-point 0 indicating the MBS ice core site. c) Proportion of HYSPLIT back-**
**trajectories associated with each cluster for all days from 1979-2016 and for days identified as an EPE from 1979-2016.**

The positive geopotential height anomaly we observe in association with EPEs suggest that snowfall
associated with EPEs often originates in the mid-latitude Southern Indian Ocean and is associated with
strong meridional transport due to atmospheric blocking (sections 3.2.1. and 3.2.3). To investigate this
using the HySPLIT back-trajectories, we filtered trajectories to only include those days identified as an



EPE (Table 3). When this filtering is applied, we see a shift in the percentage of trajectories associated with each cluster (Fig. 5c). There is a sharp decrease in the trajectories associated with the local
pathway (cluster 1) from 36.7% to 25.4% and a greater than two-fold increase in the trajectories in cluster 4 from 9.5% to 22.9%, which follows a cyclonic pathway from the Southern Indian Ocean. There is also a weaker decrease in trajectories associated with cluster 2 (easterly coastal route) and cluster 5 (trans-continental route) and an increase in cluster 3 (westerly coastal route).

| | *Percentage of trajectories* | | | | | |
|---|---|---|---|---|---|---|
| | **Cluster 1** %| **Cluster 2** %| **Cluster 3** %| **Cluster 4** %| **Cluster 5** %| **Total %** |
| | **ALL TRAJECTORIES** | | | | | |
| **All data** | 36.7 | 26.5 | 22.8 | 9.6 | 4.4 | 100 |
| *DJF* | 9.5 | 10.0 | 2.6 | 2.0 | 0.9 | 25 |
| *MAM* | 9.7 | 4.4 | 8.1 | 2.1 | 0.7 | 25 |
| *JJA* | 8.5 | 5.9 | 6.7 | 2.6 | 1.3 | 25 |
| *SON* | 9.1 | 6.3 | 5.5 | 2.6 | 1.5 | 25 |
| | **EPE TRAJECTORIES ONLY** | | | | | |
| **All data** | 25.4 | 22.2 | 27.9 | 22.9 | 1.6 | 100 |
| *DJF* | 4.1 | 5.7 | 3.6 | 5.7 | 0.2 | 19.3 |
| *MAM* | 7.0 | 5.2 | 12.4 | 4.1 | 0 | 28.8 |
| *JJA* | 7.3 | 7.5 | 6.6 | 5.9 | 1.1 | 28.4 |
| *SON* | 7.0 | 3.9 | 5.2 | 7.3 | 0.2 | 23.6 |


**Table 3: Percentage of air mass trajectories associated with each individual cluster for Mount Brown South from HySPLIT analysis (Figure 5). The upper table indicates all trajectories for the period 1979-2016 while the lower panel includes only trajectories on days identified as EPEs. For each group, we show both the total percentage of trajectories associated with each cluster for the period 1979-2016 (all data) and the percentages associated with each cluster for each season (DJF, MAM, JJA,**
**SON).**

We also find strong seasonality for some clusters when filtering for EPEs is applied. Of particular note is the strong preference for trajectories to be associated with cluster 3 during austral autumn (MAM). This cluster describes a more zonal flow to the ice core site than cluster 4, which has strong meridional flow. This may explain why we see a decreased correlation with accumulation and blocking during
autumn (section 3.2.3) as extreme precipitation during autumn may be associated with atmospheric conditions that drive zonal flow during this season.

Regardless of the differences in pathways, when we examine the geopotential height anomalies for each cluster during EPEs we still find a positive geopotential height anomaly to the east of the MBS site (Fig.
A4 in the Appendix). The strength and positioning of this anomaly varies for each cluster (with clusters





3 and 4 displaying an anomaly pattern most similar to the mean anomaly for all EPE events). This suggests that mid-latitudinal blocking is a persistent feature of EPEs regardless of the transportation pathway.

### 3.4 Atmospheric rivers

Atmospheric rivers represent a subset of EPEs. As atmospheric rivers are typically associated with strong meridional transport, warm temperature anomalies, and extreme snowfall we would expect these events to be especially impactful on mean annual accumulation and water isotope records in ice cores. Many methods have been developed to identify atmospheric rivers, however we choose here to compare to results from Wille et al., (2021) which have been specifically tuned for identification of atmospheric river events in the high southern latitudes. This method uses integrated vapour transport (vIVT) data from the Modern-Era Retrospective analysis for Research and Applications, Version 2 (MERRA-2) to identify grid-cells that are at or above the 98th percentile of all monthly vIVT values, which are classified as an atmospheric river if they extend for at least 20° latitudinally.

We found that 25.6 % of all identified EPEs were also categorised as atmospheric rivers (and conversely, 76.5% of ARs were also identified as EPEs). When the threshold for extreme events is increased to the 95th or 99th percentile, 42.4 % (95th) and 73.3 % (99th) of EPEs are also classified as atmospheric rivers. Wille et al., (2021) found that atmospheric rivers account for a similar percentage of EPEs across Antarctic (25-35% for the 90th percentile, 35-45% for the 95th and 60-70 % for the 99th).

There is also a strong association of atmospheric rivers with cluster 4 from the HySPLIT analyses (Section 3.3.1). 32.1 % of all identified atmospheric rivers at MBS are associated with cluster 4, compared with 22.6 %, 18.2 %, 25.5 % and 1.5% associated with clusters 1, 2, 3 and 5 respectively. We would expect most atmospheric rivers to be associated with cluster 4 as this best represents the meridional onshore pathway typically associated with atmospheric river transport to Antarctica.

It is interesting to note that the highest monthly accumulation at MBS occurred in December 1989 (86.4 mm WE accumulation). This anomalously high monthly accumulation was driven by two unusual atmospheric events during December 1989 that lead to both extreme precipitation and high temperatures in the region around the MBS ice core site (Turner et al., 2022). The first event, which occurred around 5th December 1989, was caused by the combined impacts of an atmospheric river with strong downslope winds. This led to both extreme snowfall and high temperatures across coastal East Antarctica (Turner et al., 2022). A second atmospheric river impacted the region on 27-28th December, again resulting in anomalously high temperatures and snowfall. Atmospheric rivers are rare across Antarctica, and the co-occurrence of these two events in December 1989 resulted in extreme accumulation for this month, observable in the RACMO2.3p2 monthly precipitation record.

While atmospheric rivers represent an important contribution to annual snowfall across Antarctica, and are frequently associated with extreme temperature anomalies, we cannot readily differentiate the impacts of atmospheric rivers from other EPEs in ice cores. Future high-resolution snow-pit sampling





may help to quantify the magnitude of the impacts of strong atmospheric river events on the water
isotope record, however in the context of this work we cannot further differentiate the impacts of
atmospheric rivers from EPEs associated with other synoptic conditions.

## 3.5 Temperature anomalies

In Antarctica, a strong inversion layer is present over much of the ice sheet meaning that surface
temperature and condensation temperature (above the inversion layer) are not always directly related
(Jouzel and Merlivat, 1984). Water isotope records are directly dependent on condensation temperatures
rather than surface air temperature. We therefore calculated the temperature anomaly relative to the 30-
day seasonal mean temperature for all EPE days identified by RACMO2.3p2 using both surface (t2m)
and condensation temperature (approximated at the 650 hPa pressure level) data from ERA-5 (see
section 2.2.2).

There is a strong positive temperature anomaly across Wilhelm II Land during EPEs at MBS (Fig. 4b).
In the surface temperature field, there is a strong anomaly centred over the MBS site, with a mean
positive anomaly of +7.0° C for the grid cell containing the ice core site. This warm anomaly also
extends over a broad region of East Antarctica, from 60° E-105° E, and inland to 80° S. As the pressure
level representative of condensation varies across the region, we only calculated the condensation
temperature for the grid cell containing the MBS site. The condensation temperature anomaly during
EPEs is weaker but still present, with an anomaly +2.5° C for the MBS grid cell. The temperature
anomalies in both surface and condensation temperature fields are again likely associated with
atmospheric blocking, which causes increased meridional flow of warm, moist air from the mid-
latitudes to the MBS site during EPEs. This results in both positive temperature anomalies and high
accumulation.

### 3.5.1 Temperature bias

As accumulation from EPEs makes up nearly 50% of annual snowfall at MBS, the impact of warm
temperature anomalies during these events would be expected to generate a warm bias in the ice core
record. To investigate this impact, we calculated precipitation-weighted temperature records, based on
daily accumulation amounts from RACMO2.3p2 and daily mean condensation temperatures
(approximated as 650 hPa temperature) from ERA-5. We generated both a monthly precipitation-
weighted temperature record, which calculates the mean temperature for each month weighted to the
amount of precipitation on each day, and an annual precipitation-weighted temperature record, which
calculates a mean annual temperature using the same method but for an entire year, therefore also
accounting for the winter bias in accumulation.

Monthly precipitation-weighted temperatures were calculated as:

$$T_{\text{pr-monthly}} = \frac{\sum_{\text{day}=1}^{31} T_{\text{day}} \times Pr_{\text{day}}}{\sum_{\text{day}=1}^{31} Pr_{\text{day}}} \qquad (1)$$

where $T_{\text{day}}$ is the daily temperature and $Pr_{\text{day}}$ is the daily accumulation.




Annual precipitation-weighted temperatures were calculated as:

$$T_{pr\text{-}annual} = \frac{\sum_{day=1}^{365} T_{day} \times Pr_{day}}{\sum_{day=1}^{365} Pr_{day}} \qquad (2)$$

When the temperature bias is examined at the monthly scale (the mean monthly precipitation-weighted temperature from 1979-2016 compared with the mean monthly temperature for the same period), we

find that there is a mean positive temperature bias of $0.90 \pm 0.40°$ C (Fig. 6d). This temperature bias is strongest during the winter months (mean anomaly from May-September = $1.21 \pm 1.51°$ C), coincident with the highest occurrence of EPEs, and the lowest during austral summer (mean anomaly Dec-Jan = $0.19 \pm 1.32°$ C) when there are fewer EPEs (Fig. 6b and 6d).

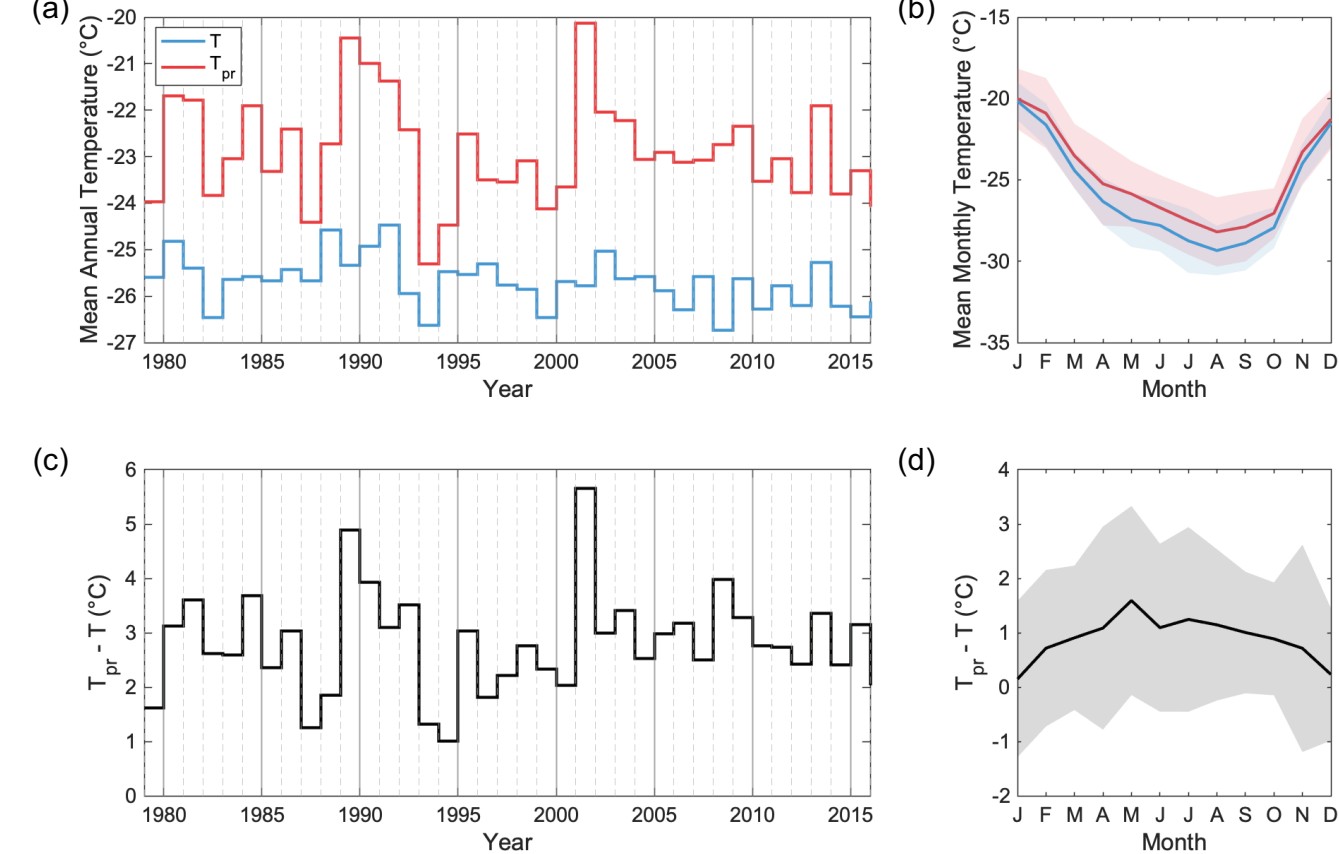


Figure 6: a) Mean annual condensation temperature (approximated as 650 hPa temperature) from ERA-5 for 1979-2016 at MBS (solid blue line)) and the mean annual precipitation-weighted condensation temperature for the same period (solid red line). b) Mean monthly condensation temperature from 1979-2016 (solid blue line, with shading representing one standard deviation) and mean monthly precipitation-weighted temperatures for the same period (solid red line, with shading representing one standard

deviation). c) Temperature difference between the mean annual precipitation-weighted temperature and mean annual temperature. d) Temperature difference between the mean monthly precipitation-weighted temperature and the mean monthly temperatures from 1979-2016, with shading representing one standard deviation.





At the annual scale, the mean warm bias is 2.8 ± 0.9° C but there is considerable inter-annual variability (Fig. 6a and 6c). The largest warm bias occurs in 2001, where the precipitation-weighted mean annual temperature is 5.7° C warmer than the mean annual temperature. Conversely, in 1994 the temperature bias is only 1.0° C. The large warm bias in 2001 is also associated with the highest occurrence of EPEs in a year (34), while the small warm bias in 1994 is associated with the fewest EPEs in a year (7). There

is a significant positive correlation between the temperature bias and number of EPEs in a year (r= 0.641, p < 0.001; Fig. A5 in the Appendix).

**3.6 Mount Brown South water isotope record**

**3.6.1 Water isotope terminology**

We present the water isotope record for 4 - 20 m of the MBS-Main (1979 - 2009) and 0 - 20 m of the MBS-Charlie (1979 – 2018) ice cores (Fig. 7a and b). The ice core water isotope record is a reflection of the conditions present during deposition, and will thus be biased towards recording the climate

conditions associated with extreme precipitation at MBS. Here we discuss the impacts of precipitation intermittency on the $\delta^{18}O$ record in the MBS ice core, as well as the secondary parameter, deuterium-excess. We choose to present the deuterium-excess parameter as $d_{ln}$, defined by Uemura et al., (2012) as:

$d_{ln} = \delta'D - (A \times (\delta'^{18}O)^2 + B \times \delta'^{18}O)$         (3)

where the constants of A = -28.5 and B = 8.47, and $\delta'^{18}O$ and $\delta'D$ refer to $\delta'_x = \ln(1 + \delta_x)$.

This definition accounts for the inherent non-linearity in the relationship between $\delta^{18}O$ and $\delta D$ and

reduces biases caused by an assumed linear relationship (as assumed in the traditional measure of deuterium-excess: $d_{excess} = \delta D - 8 \times \delta^{18}O$). The causes of these biases and the argument for the adoption of $d_{ln}$ in ice core research has previously been discussed in detail (Uemura et al., 2012; Markle et al., 2016). These biases have been demonstrated to be more problematic at sites further inland than MBS, where water masses have undergone further kinetic fractionation due to longer transportation pathways

and greater temperature gradients. In practice we see a very close agreement between $d_{ln}$ and $d_{excess}$ measurements at the MBS site.



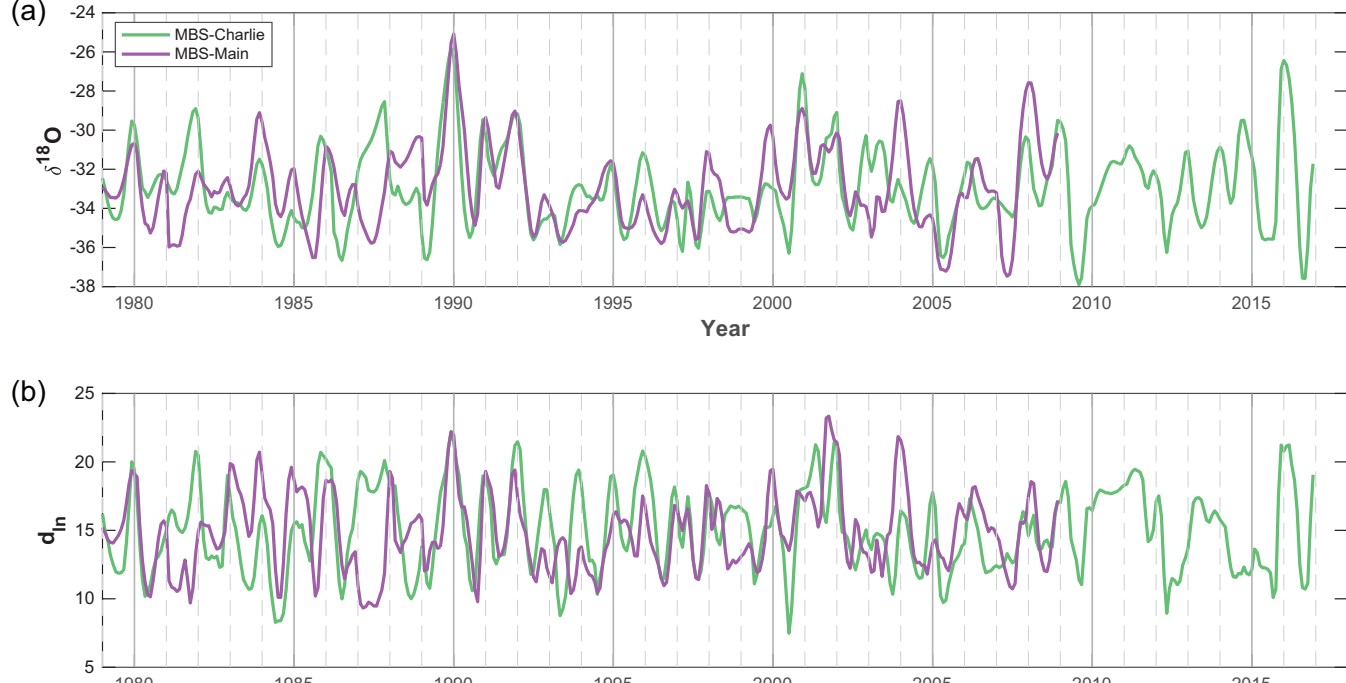

Figure 7: a) Ice core $\delta^{18}O$ profiles and b) ice core $d_{ln}$ profiles from MBS-Main for 1979-2008 (purple) and MBS-Charlie for 1979-2016 (green).

### 3.6.2 Correlation with temperature

To investigate the relationship between temperature and the water isotope records from both MBS-Main and MBS-Charlie, we compared the annually-averaged (January - December) $\delta^{18}O$ record with condensation temperatures (650 hPa temperatures) from ERA-5, and the annual precipitation-weighted temperature record derived ERA-5 condensation temperatures weighted by RACMO2.3p2 daily precipitation amounts (Section 3.5.1).


For MBS-Main, there is no significant correlation between mean annual $\delta^{18}O$ and condensation temperature (Fig. 8a). However, a strong positive correlation is found when the precipitation-weighted temperature is considered instead (r = 0.527, p=0.003; Fig. 8b). For MBS-Charlie, the results are more complex. When the full record is considered (1979-2016), there is no significant correlation between the
mean annual $\delta^{18}O$ and condensation temperature (r=0.200, p=0.228) nor precipitation-weighted temperatures (r=0.294, p=0.073). When only a partial record is considered (1979-2008, the same period covered by MBS-Main) the correlation improves but is still insignificant between $\delta^{18}O$ and condensation temperature (r=0.305, p=0.101). However, a significant positive relationship is found between $\delta^{18}O$ and the precipitation-weighted temperature for the shorter time period (0.432, p=0.017;
Table 4). This suggests that there may be non-stationarity in the relationship between $\delta^{18}O$ and the precipitation-weighted temperature at this site, or that errors in dating in the uppermost section of the



MBS-Charlie core have resulted in an offset between the isotope record and the temperature record
(Section 2.1.3). From 2009-2012 there is very weak cyclicity in the ice core $\delta^{18}O$ record which may
potentially contribute to dating errors in the upper section of the core, thereby impacting the apparent
correlation between $\delta^{18}O$ and temperature (Fig. 7a).

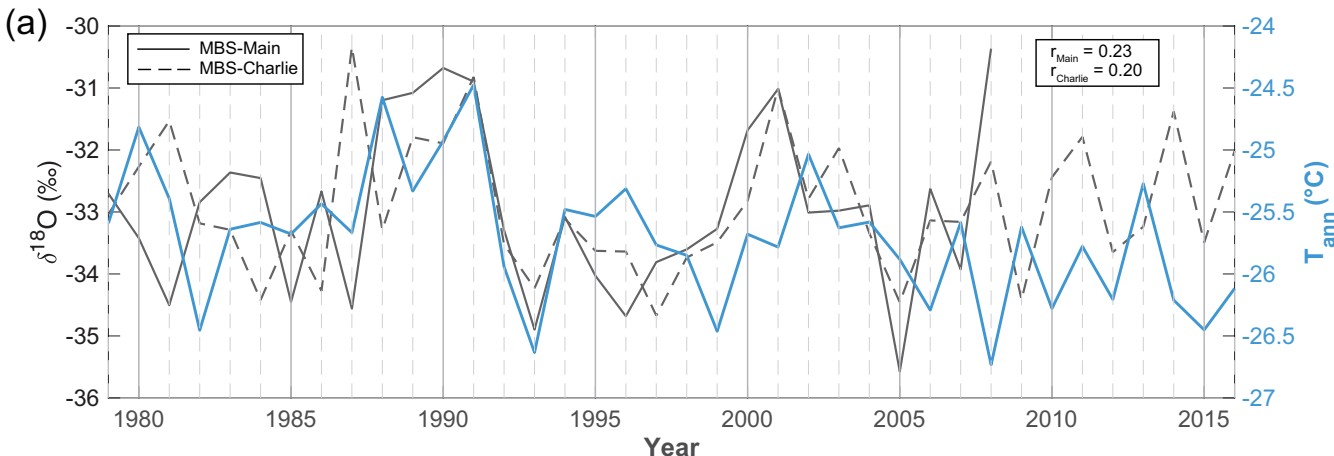

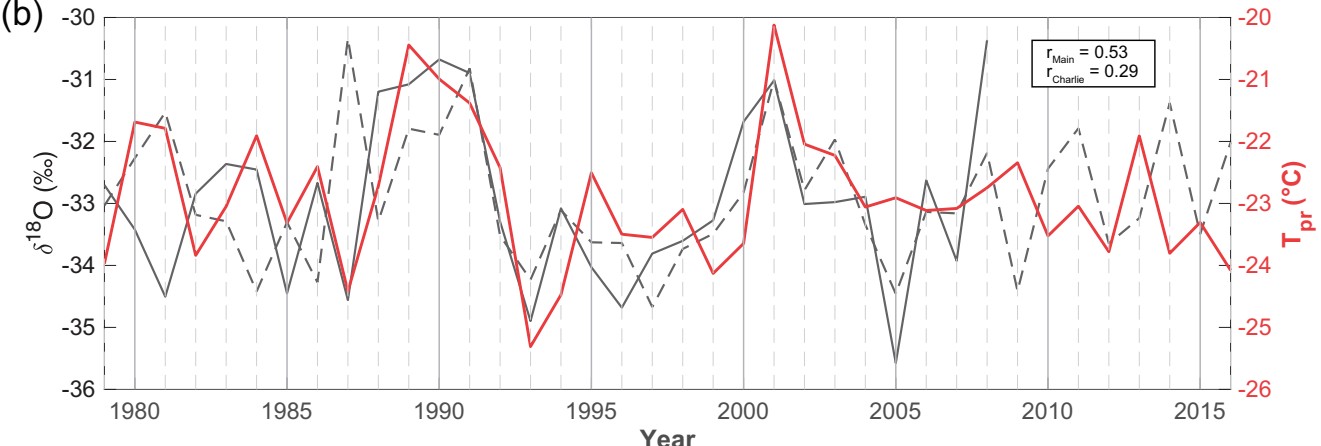

**Figure 8: a) Annually-averaged $\delta^{18}O$ from MBS-Main (solid black line) and MBS-Charlie (dashed black line) and condensation**
**temperature from ERA-5 from 1979-2016 (solid blue line). b) As in a, but with ice core $\delta^{18}O$ data shown alongside precipitation-**
**weighted condensation temperature from ERA-5 for 1979-2016 (solid red line).**

$d_{ln}$ in both MBS-Main and MBS-Charlie has a weaker correlation with both the condensation
temperature and the precipitation-weighted temperature (Table 4). A significant correlation is only seen
between $d_{ln}$ and the precipitation-weighted temperature record in MBS-Main (r=0.484, p=0.007).
Unlike $\delta^{18}O$, $d_{ln}$ is primarily impacted by kinetic fractionation processes that occur during evaporation
rather than site conditions, and so we would not expect there to be strong correlations between $d_{ln}$ and
condensation temperatures.




| | Time period | Condensation Temperature | | Precipitation-weighted condensation temperature | |
|---|---|---|---|---|---|
| | | r | p | r | p |
| **δ¹⁸O** | | | | | |
| MBS-Main | 1979-2008 | 0.226 | 0.231 | **0.527** | **0.003** |
| MBS-Charlie | 1979-2016 | 0.200 | 0.228 | 0.294 | 0.073 |
| MBS-Charlie | 1979-2008 | 0.305 | 0.101 | **0.432** | **0.017** |
| **$d_{ln}$** | | | | | |
| MBS-Main | 1979-2008 | 0.111 | 0.559 | **0.484** | **0.007** |
| MBS-Charlie | 1979-2016 | 0.075 | 0.653 | 0.199 | 0.231 |
| MBS-Charlie | 1979-2008 | 0.043 | 0.820 | 0.219 | 0.246 |

**Table 4: P Pearson's correlation coefficient (r) and p-values for annually averaged δ¹⁸O and $d_{ln}$ values from MBS-Main and MBS-Charlie, for for the full record (1979-2016) and a shorter record (1979-2008; see text for details), with condensation temperatures**
**and precipitation-weighted condensation temperatures. Significant correlations (p < 0.05) are shown in bold.**

There are a number of estimates of the δ¹⁸O-T slope from across Antarctica. The continental-scale spatial slope has been calculated from variations in surface snow δ¹⁸O and site mean annual temperatures, to calculate a δ¹⁸O-T slope of 0.75-0.80‰ °C⁻¹ (Masson-Delmotte et al., 2008).
Alternatively, temporal slopes for individual locations (based on seasonal variations of δ¹⁸O in surface snow and temperature through time at a single location) have been calculated at many sites across Antarctica (e.g. Stenni et al., 2016 and references therein). The temporal δ¹⁸O-T slopes are typically about half of the spatial slope, although individual slopes are site specific and display considerable variability.


We calculate a δ¹⁸O-T slope for MBS-Main of 0.57 ± 0.12 ‰ °C⁻¹ and for MBS-Charlie of 0.73 ± 0.14 ‰ °C⁻¹, using a geometric mean regression. The slopes are calculated using mean annual δ¹⁸O and mean annual precipitation-weighted temperatures, due to the stronger relationship between δ¹⁸O and the precipitation-weighted temperature than mean annual temperature. The geometric mean regression
method is preferred over a least squares regression as it assumes uncertainty in both the y-dimension (δ¹⁸O) and the x-dimension (T), unlike a least squares method n which assumes uncertainty only in the y-dimension. Given the inherent uncertainties around the temperature against which we are calibrating due to a lack of direct of measurements of site temperature during deposition, the geometric mean regression method was chosen as a more appropriate method for calculating the temporal slope. Using




the least squares regression will underestimate the sensitivity of $\delta^{18}$O to T, which results in a reduced
$\delta^{18}$O-T slope for both MBS-Main (0.24 ± 0.10 ‰ °C$^{-1}$) and MBS-Charlie (0.27 ± 0.11 ‰ °C$^{-1}$).

The calculated $\delta^{18}$O-T slopes are comparable to other temporal slopes across Antarctica. A slope of 0.44
‰ °C$^{-1}$ was calculated for the Law Dome ice core (by comparing the mean isotope seasonal cycle from
1304-1988 to mean monthly temperatures from 1986-1991; van Ommen and Morgan, 1997) ). Stenni et
al., (2016) calculated a mean temporal slope of 0.47 ‰ °C$^{-1}$ for East Antarctica based on calculations
from many individual studies that utilised a variety of methods including precipitation samples, firn
sampling and ice core sampling compared with both direct and re-analysis temperature measurements.
We note that an accurate estimation of the temporal slope at MBS would require daily snow sampling
and direct temperature observations from the site, but despite limitations in the methodology we obtain
comparable results to other more highly resolved surface snow and firn-core records.

We note that the maximum $\delta^{18}$O values in both MBS-Main and MBS-Charlie occurred during
December 1989/January 1990 (Fig. 7a). As discussed earlier (Section 3.4), December 1989 was an
anomalous month for both accumulation and temperatures in the MBS region (Turner et al., 2022). The
co-occurrence of two large atmospheric river events lead to both extreme temperatures and
accumulation in the region around the ice core site. Detailed isotope-enabled modelling of these events
would be required to quantify the impacts of the atmospheric rivers on the water isotope record, which
is beyond the scope of this study. However, we would expect high $\delta^{18}$O values to be associated with
these large atmospheric river events due to reduced Rayleigh fractionation from the source region to the
site caused by the rapid transport and reduced temperature gradient during these events.

## 4. Conclusions

This study has used a combination of ice core data, re-analysis products, and models to understand how
precipitation intermittency impacts the temperature records preserved in an East Antarctic ice core.
Accumulation at the Mount Brown South ice core site is not constant and continuous, but instead shows
clear seasonality and interannual variability. Much of this variability can be explained by variability in
extreme precipitation events, or days where daily accumulation exceeds 2.7 mm WE day$^{-1}$. There is a
greater occurrence of extreme accumulation events during the austral winter months (May-June), which
also coincides with the highest monthly mean accumulation. Much of the inter-annual variability in
accumulation is also caused by variability in EPEs. We found that years with a greater number of EPEs
also have higher annual accumulation than those with fewer events. Although EPEs only occur on
average on 6.3 % days each year, they account for 43.5 % of annual snowfall, and so the synoptic
conditions associated with extreme snowfall are over-represented in the MBS ice core.

We found that extreme events tend to be associated with strong meridional transport driven by large
blocking highs to the east of the MBS ice core site, which can be identified in mean 500 hPa
geopotential heights. Precipitation during the summer months, both from extreme events and total
accumulation is positively correlated with atmospheric blocking. During the winter months, there is a
weaker association between extreme events and blocking, however blocking still plays an important





role in the total accumulation. Investigation into back-trajectories associated with extreme events using HySPLIT further confirms that precipitation during these events is more frequently associated with direct transport from the mid-latitude Indian Ocean to the ice core site.


The increase in meridional flow during extreme events is associated with direct transport of warmer air masses from the mid-latitudes to the ice core site, meaning that extreme precipitation is also associated with a strong positive temperature anomaly. We demonstrated that there is a mean surface temperature anomaly of +7 °C during EPEs at the MBS site and a +2.5 °C anomaly above the inversion layer. Water

isotopic records in ice cores are principally an archive of the climate conditions during precipitation. A consequence of this is that the mean annual isotopic composition of the ice core records is not representative of a mean annual temperature, but instead reflects the mean annual temperature during snowfall events. We found that there is a mean warm bias of 2.8 °C when we weight the temperature record by daily precipitation amounts, although there is both considerable inter-annual and seasonal

variability in the magnitude of this bias.

The water isotope record in both MBS-Main and MBS-Charlie is not significantly correlated to mean annual condensation temperatures at the MBS site. However, $\delta^{18}O$ in both MBS-Main and MBS-Charlie is strongly correlated with the precipitation-weighted temperature record for the period from 1979-2008.

This strong correlation provides a framework for examining past climate variability at this location using the water isotope record from MBS-Main. However, untangling the competing effects of precipitation intermittency and temperature on the water isotope record is challenging. Combing the annual accumulation record from MBS with water isotope measurements alongside modelling approaches will likely improve our understanding of how these parameters can be best used to

reconstruct climate variability both at the MBS site and in the broader Southern Ocean region.

Extreme snowfall events are frequently associated with trajectories originating in the Southern Indian Ocean. This region is of particular interest due to the strong teleconnections with Australian hydroclimate as well as emerging evidence for potential climate-driven changes in the East Antarctic

Ice Sheet (Stokes et al., 2022). However, long-term records of the climate variability in this region are sparse. A long-term climate reconstruction using the full MBS isotope record will be invaluable to improving understanding of the natural climate variability in the region. This includes the second-order water isotope parameter ($d_{ln}$), which is a more direct tracer of source region conditions, and may preserve a signal of climate variability in an ocean region that is largely under-represented in long-term

climate reconstructions. A future mechanistic study into the response of $d_{ln}$ to both seasonal variability at the ice core site and seasonal changes in temperature and relative humidity in the Southern Indian Ocean would be beneficial for building a quantitative climate reconstruction from the water isotope record at the MBS site.

The full MBS ice core record, covering more than 1000 years, will be a valuable addition to global research. One of the key priorities in ice core research is to expand and develop current networks of temperature and hydroclimate records for the Common Era. Improved spatial distribution of these




records will allow for regionally-specific detail and processes to be reconstructed, giving greater context to global climate changes.

# 790  Appendix: Climatology of the Mount Brown South ice core site in East Antarctica: implications for the interpretation of a water isotope record

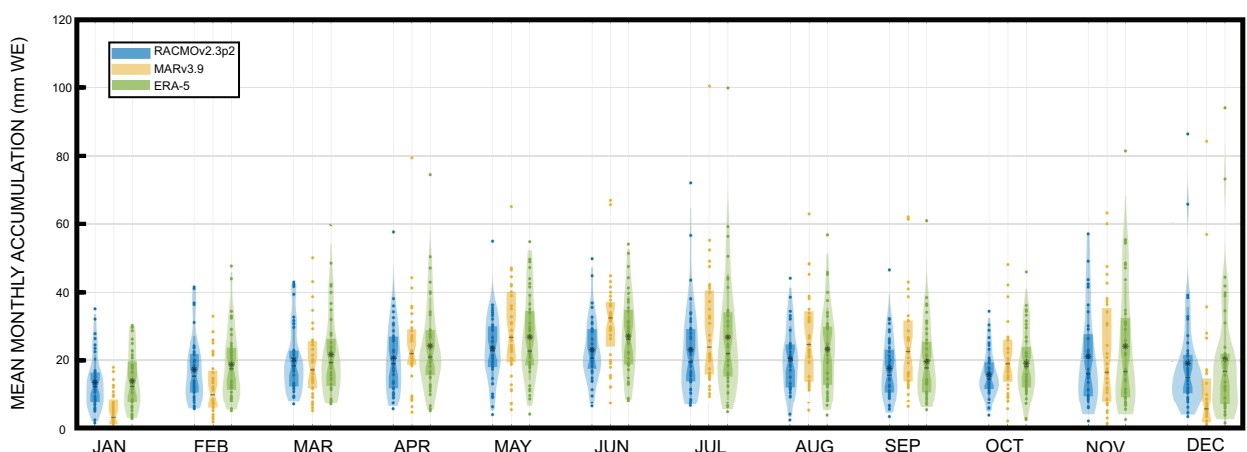

**Figure A1: Violin plots showing monthly accumulation from RACMO2.3p2 1979-2016 (blue), MARv3.9 from 1981-2018 (yellow) and ERA5 from 1979-2020 (green). Mean monthly values are indicated by an Asterix.**

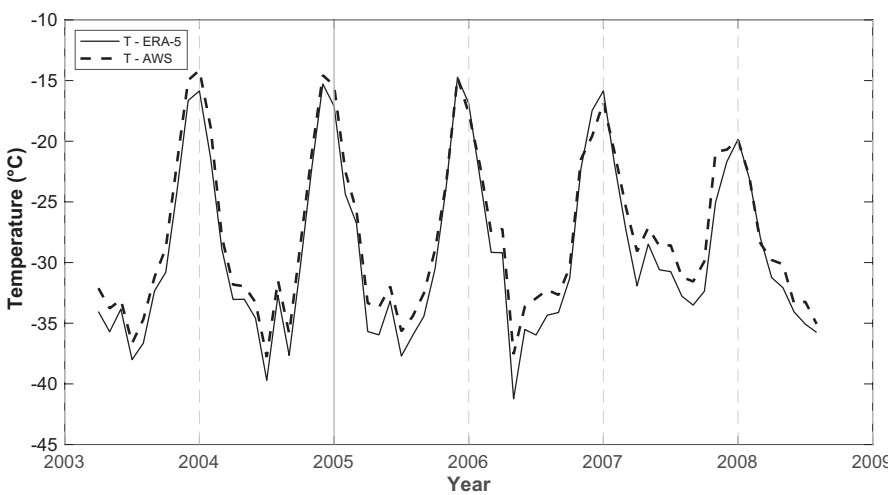

**Figure A2: Comparison of mean monthly temperatures from an automated weather station located on Mount Brown, 19 km from**
**the MBS drill site (dashed line) and mean monthly surface temperatures for the grid cell encompassing the MBS drill site from ERA-5 (solid line).**





**Figure A3: Mean 500 hPa geopotential height anomaly for all EPE days identified using RACMO2.3p2 from 1979-2016 for each season (DJF, MAM, JJA, SON) relative to the mean from ERA-5. Grey bars indicate the regions between which the blocking index is calculated.**



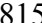


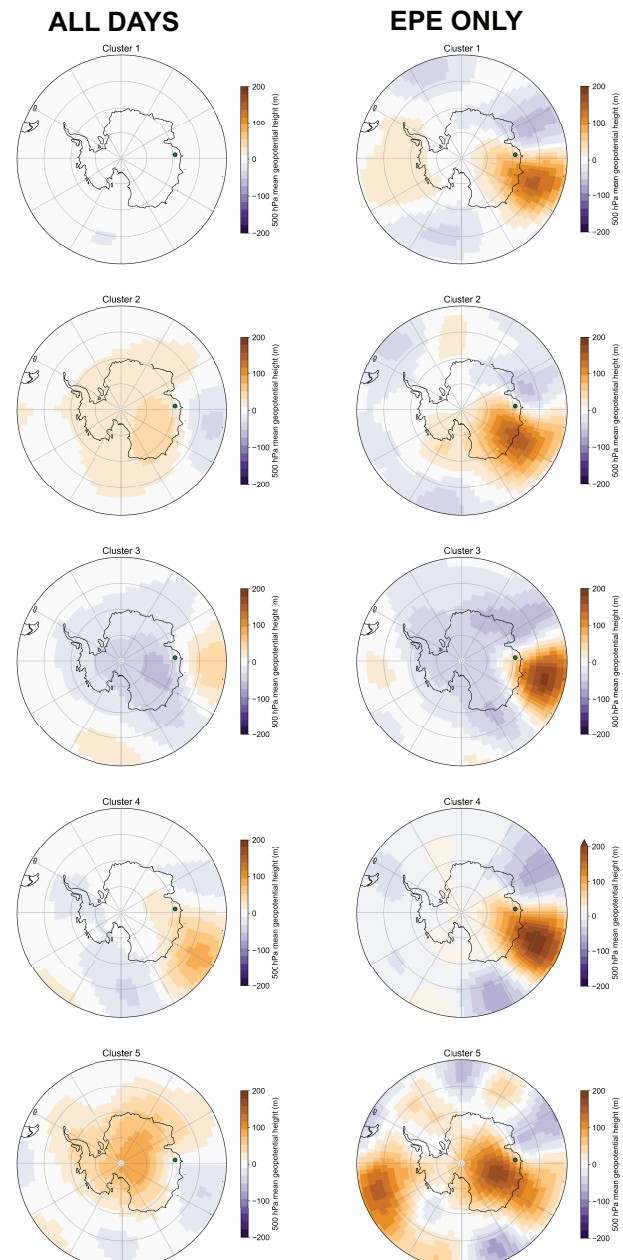

**Figure A4: Mean 500 hPa geopotential height anomaly using data from NCEP/NCAR for each cluster identified by HySPLIT back-trajectory analysis. The left column shows mean anomalies for all days in each cluster, while the right shows the anomalies only for days identified as EPEs using RACMO2.3p2 from 1979-2016.**




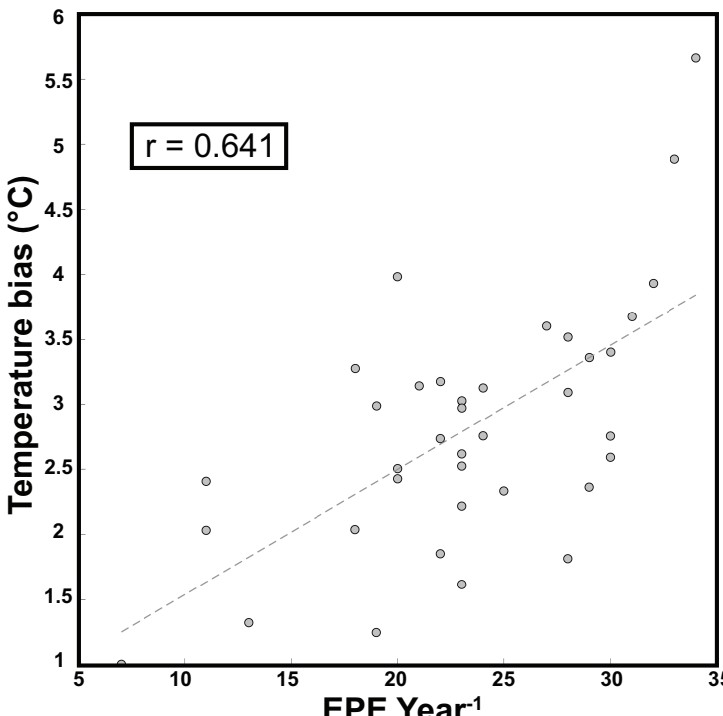

**Figure A5: Correlation between number of EPE days year$^{-1}$ and the temperature bias ($T_{pr} - T$).**


|  | RACMO2.3p2 | MARv3.9 | ERA-5 |
|---|---|---|---|
| **RACMO2.3p2** |  | 0.87 | 0.95 |
| **MARv3.9** |  |  | 0.89 |

**Table A1: Pearson's Correlation Coefficient (r) for different accumulation (RACMO2.3p2 and ERA-5) and surface mass balance products (MARv3.9).**

| EPE threshold | # of days | Self-Organising Maps Synoptic Type | | | | | | | | |
|---|---|---|---|---|---|---|---|---|---|---|
|  |  | SOM1 | SOM2 | SOM3 | SOM4 | SOM5 | SOM6 | SOM7 | SOM8 | SOM9 |
| 90th | 897 | 34.4 % | 11.3 % | 15.2 % | 8.8 % | 7.0 % | 2.9 % | 11.5 % | 2.0 % | 6.9 % |
| 95th | 449 | 41.0 % | 8.7 % | 13.1 % | 7.8 % | 6.7 % | 2.9 % | 11.6 % | 0.9 % | 7.3 % |
| 99th | 90 | 53.3 % | 4.4 % | 20.0 % | 3.3 % | 5.6 % | 0 % | 6.7 % | 0 % | 6.7 % |

**Table A2: Percentage of days associated with each synoptic type from 1979-2016 from Udy et al., 2021. Days are filtered to only**

**include days where an extreme precipitation day is identified. The threshold for EPE events is set at the 90th, 95th and 99th percentile.**



**Data availability:**

The datasets used in this study are available online from the following locations: Temperature and
geopotential height data from ERA-5: https://apps.ecmwf.int/datasets/. Geopotential height data from
NCEP/NCAR:
http://www.psl.noaa.gov/data/gridded/data.ncep.reanalysis.html. Daily precipitation amounts from
RACMO2.3p2 https://doi.org/10/c2pv. Surface mass balance from MARv3.9:
https://zenodo.org/record/5195636#.Y1dtN-xByAk. Mount Brown South Automated Weather Station
temperature data: https://data.aad.gov.au/metadata/records/antarctic_aws. Atmospheric River catalogue:
https://www.earthsystemgrid.org/dataset/ucar.cgd.artmip.html. The daily synoptic typing dataset for the
Southern Indian Ocean:
https://data.aad.gov.au/metadata/records/AAS_4537_z500_SynopticTyping_SouthernIndianOcean.
Mount Brown South ice core records accumulation data and dating:
https://data.aad.gov.au/metadata/records/AAS_4414_MountBrownSouth_LawDome_icecores_seasalt_a
ccumulation_2020

**Author contributions:**
SLJ led the study, including the data analysis and writing of the manuscript. SLJ and NJA conceived the
concept. TRV and NJA provided funding acquisition and assisted with project administration. TRV,
CKC, ADM and CTP contributed to ice core analysis and dating. All authors contributed to writing the
manuscript and declare that they have no conflict of interest.

**Competing interests:**
The authors declare that they have no conflict of interest.

**Acknowledgements:**
We thank all involved with the drilling and collection of the Mount Brown South ice core (Sharon
Labudda, Paul Vallelonga, Allison Criscitiello, Jason Roberts, Peter Campbell). Sarah L. Jackson is
supported by an Australian Research Training scholarship. Sarah L. Jackson and Nerilie. J. Abram are
both supported by the Australian Research Council (ARC) Centre of Excellence for Climate Extremes
(CE170100023) and the ARC Australian Centre for Excellence in Antarctic Science (SR200100008).
Tessa R. Vance acknowledges support from an ARC Discovery Project (DP180102522; DP220100606)
as well as the ARC Special Research Initiative for Antarctic Gateway Partnership (SR140300001) and
the Australian Antarctic Program Partnership (ASCI000002). This work contributes to Australian
Antarctic Science projects (4414, 4537, and 4061), and a National Science Foundation project (NSF
P2C2 18041212).



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
