# Peer review of "Climatology of the Mount Brown South ice core site in East Antarctica: implications for the interpretation of a water isotope record"

_EGUsphere, 2022_

## Referee Comment (RC2)

**Review of Climatology of the Mount Brown South ice core site in East Antarctica: implications for the interpretation of a water isotope record**

The manuscript describes the climatic conditions at the Mount Brown South ice core drilling site in Antarctica, and the implications for the interpretation of the isotopic composition as a paleoclimatic record. The manuscript combines observations, reanalysis, and back-trajectories analyses to evaluate the contribution of warm synoptic events to the signal. Specifically, the authors address the global contribution of extreme precipitation events, a subset of events which accounts for the largest total precipitation, which includes atmospheric rivers. The results are compared to the upper part of the main ice core from Mount Brown South, as well as from a short replicate core.

Overall, such studies evaluating the impact of the local climatic conditions to the isotopic signal stored in the ice core should be necessary preamble approaches to the interpretation of the isotopic paleothermometer. The authors combine a large range of polar meteorology tools to constrain the impact of precipitation intermittency and seasonality which are two limiting factors to the interpretation of the water isotopes at high resolution. The methodology is appropriate; the figures convey the necessary information. Overall, the manuscript is well written but I believe the following points should be taken into account before it is considered for publication.

**General comments:**

I acknowledge that this is already a very dense manuscript, which includes results ranging across disciplines and provide an in-depth study of the impact of the signal of the largest precipitation events imprinted in the ice core. In my opinion, the manuscript presently falls just short of achieving its goal, i.e. "understand how precipitation intermittency impacts the temperature records preserved in an East Antarctic ice core" (first sentence of the conclusion), because the pertinence of the results for signal analysis is not sufficiently discussed. To move beyond *illustrating* the impacts of precipitation intermittency on the signal, and really *understand* them, I would suggest including the following discussion elements:

- While it's true that precipitation intermittency, and in particular the fact that synoptic events are associated with warmer than conditions (and thus a warm offset), for the interpretation of the isotopic records, which are always given as an anomaly, it does not matter. The most important question is for each time scale, if the amplitude of the variations of isotopic composition and of temperature can be scaled by the same factor. Section 3.6.2 which evaluates the correlation between isotopic composition and temperature is disconnected from the rest of the studies which address the EPE influence. At this point, new calculations evaluating the modelled isotopic signal with and without the contribution of the EPEs might be out of the framework of the manuscript, but discussing the influence of these large events, building up on the results from (Sime et al., 2009), (Casado et al., 2020), and (Münch et al., 2021) would support the main goal of the manuscript,
- The impact of stratigraphic noise is completely ignored (Ekaykin et al., 2004; Fisher et al., 1985; Münch et al., 2016; Petit et al., 1982), and since it is similar than precipitation intermittency, it needs to be at least discussed. Specifically, here, the distance between the cores will be key to evaluate if the stratigraphic noise will affect the two cores the same way, or not, both in term of dating uncertainty, and of noise added to the isotopic signal.
- Overall, the use of the condensation temperature here is not based on physics, since the temperature time series are simply compared with the isotopic records. Since the condensation temperature is an artificial temperature retrieved from the model at a variable location, which is solely based on the pressure level, and not on where the model is predicting that the precipitation takes place, it does not bring any added value to you. How are the results of Section 3.6 for precipitation weighted surface temperature? While I agree that in principle, the fractionation takes place way above the surface when the moisture condenses, the signal is actually only weakly affected by the local fractionation coefficient sensitivity to temperature when you actually calculate a Rayleigh distillation, and the link between temperature and isotopic composition is acquired by integrating the whole distillation (Rayleigh, 1902).
- At the end, section 3.6 is disconnected from the section 3.1 to 3.5, even though, this is the important aspect. How much of the results of section 3.6 can you explain from the EPE? Can you compare the

temperature record to the temperature record with precipitation intermittency, and also with precipitation intermittency but excluding the EPE events? What does it teach you for the interpretation of the d18O of the MBS core?

**Specific comments:**

Line 59: The following studies should be cited here (Sime et al., 2009), (Casado et al., 2020), and (Münch et al., 2021).

Lines 64-65, and then later on, 83-84: the percentiles at which the EPE are defined should be consistent throughout the manuscript. Either, it needs to be based on other studies (such as the Wille et al, 2021), and then defined just once in the introduction, or, if you chose to apply your own threshold, it needs to be defined in the methods, sensitivity tests need to be applied (more cumbersome).

Lines 162 – 164: Is $^{17}$O-excess actually used in this paper? If it isn't this is unnecessary information to add here.

Section 2.2: It is not clear to me why the EPEs are characterised with RACMO and the temperatures, geopotential,… are extracted from ERA5. The dataset (RACMO, ERA5, or other) should here be only a tool used to apply the method (such as the routine from Turner et al, 2019 to identify EPEs), and ideally, it should be reproducible with both tools. Using both datasets as a gridded meteorological data input is not necessarily bad, if both are used completely for all diagnoses, and if both datasets are used for sensitivity tests, but here, it raises the question whether the results can be reproduced using one single dataset only. I believe that a coherent ensemble is more important than "direct comparisons with previous work".

Lines 259 to 261: "Here, we use HySPLIT to generate 5-day back-trajectories (120 hours), originating at the MBS site at a height of 1500 m above ground level, which is equivalent to approximately 3500 m above sea level."

Here, you evaluate the formation of precipitation at 1500 m. agl. Later on, the condensation temperature is defined as the 650mbar pressure level (line 554). Is this coherent?

Line 317 (and several times later on): if the error bar is larger than one, the '.2' is not a significant figure.

Line 549 to 555: "In Antarctica, a strong inversion layer is present over much of the ice sheet meaning that surface temperature and condensation temperature (above the inversion layer) are not always directly related (Jouzel and Merlivat, 1984). Water isotope records are directly dependent on condensation temperatures rather than surface air temperature. We therefore calculated the temperature anomaly relative to the 30-day seasonal mean temperature for all EPE days identified by RACMO2.3p2 using both surface (t2m) and condensation temperature (approximated at the 650 hPa pressure level) data from ERA-5 (see section 2.2.2).".

This is not exactly true. The local fractionation coefficient is related to the local condensation temperature (Jouzel and Merlivat, 1984), not the water isotopic composition which integrates all the successive fractionation coefficients throughout the Rayleigh distillation (Ciais and Jouzel, 1994; Rayleigh, 1902; Schoenemann et al., 2014). As a result, the most important factor is the rainout fraction (which follows Clausius Clapeyron's law, and thus, is a logarithmic ratio of the local to source temperature). Overall, the fractionation coefficient variation with temperature is only a second order parameter (Bailey et al., 2019). Taking into account the surface temperature or the condensation temperature is then just a convention at this point.

Section 3.5.1: The warm bias induced by the correlation between strong precipitation events and temperature was already used in (Sime et al., 2009), (Persson et al., 2011), and (Casado et al., 2018). The proposed strategy could help enhanced the discussion. Since the purpose is to apply these results to the interpretation of the isotopic signal, it would make sense to evaluate if this is only going to be a bias

without any frequency dependency or if this create a signal that affects particularly the high frequency variability of the signal. A bias does not matter for the interpretation of the paleothermometer because only the anomalies are studied.

Lines 622: The use of deuterium excess here appears superfluous and raises more questions than answers: (i) why is the dexcess introduced here, near the end of the Results and Discussion sections, and not in the introduction or in the methods ? (ii) why is the dexcess introduced at all since it's not used in the manuscript ? (the correlation will temperature will be discussed later on) (iii) the non-linearity of the relationship between d18O and dD will not appear on a single site with small variations of temperature, so why is the manuscript using a non-classical definition of dexcess ? (iv) and, in turn, what is the classical d-excess definition looking like? Both definitions have positive and negative aspects, but here, it is not clear what is the benefit of using this definition. Overall, I would recommend to remove the dexcess completely from the manuscript, and focus more on the link between d18O and accumulation, but if you decide to keep the dexcess, it would be necessary to explain why this specific definition is used here, even though, the classical definition has been shown "be more problematic at sites further inland than MBS", and also (and maybe more importantly), at much larger time scales which the variations of temperature were much larger.

Lines 640: There is no discussion on the stratigraphic noise, and if this could explain why the two cores have weak correlation.

Lines 665: Dating ice cores using the water isotopes has been already proven to lead to high errors when the signal to noise ratio is low (Laepple et al., 2018), which could be discussed here.

Lines 671 to 677: What is the conclusion from the piece of information? The unexpected correlation between dexcess and temperature could be link with three possible explanations: (i) the definition of dexcess is not adapted and a lot of the 1$^{st}$ order (d18O) signal remains in the dataset, (ii) it is actually a source temperature signal, but condensation and source temperature are correlated and this explains this link, or (iii), this is a complete random correlation (according to the pvalue, there is 0.7% chance of this happening).

Since there is a strong positive correlation between the dln and d18O in figure 7, it seems likely that the use of dexcess here (using either the classical or log definition) is not bringing additional information, but this should be tested if the dexcess is kept in the manuscript.

Line 696 to 697: The error bars appear very small considering the relatively low coefficient correlation (Higbie, 1991), I would suggest rather 0.2 and 0.3 respectively. A simple test can be to compare the slope of d18O with the temperature to the inverse of the slope of the temperature to d18O (which should be equal normally), but sometimes offer difference which can be used to evaluate the error bars.

Lines 696 compared to the previous paragraph: it seems here that you are comparing the condensation temperature to isotope slopes from your results to mostly surface temperature datasets. How is this affecting your results? Does using the condensation temperature really help the study? Overall, the compared data should be coherent, especially because variations of surface temperature are not scaled one to one with condensation temperature.

Bibliography

Bailey, A., Singh, H. K. A. and Nusbaumer, J.: Evaluating a Moist Isentropic Framework for Poleward Moisture Transport: Implications for Water Isotopes Over Antarctica, Geophys. Res. Lett., 46(13), 7819–7827, doi:https://doi.org/10.1029/2019GL082965, 2019.

Casado, M., Landais, A., Picard, G., Münch, T., Laepple, T., Stenni, B., Dreossi, G., Ekaykin, A., Arnaud, L., Genthon, C., Touzeau, A., Masson-Delmotte, V. and Jouzel, J.: Archival processes of the water stable isotope signal in East Antarctic ice cores, Cryosph., 12(5), 1745–1766, doi:10.5194/tc-12-1745-2018, 2018.

Casado, M., Münch, T. and Laepple, T.: Climatic information archived in ice cores: impact of intermittency and diffusion on the recorded isotopic signal in Antarctica, Clim. Past, 16(4), 1581–1598, doi:10.5194/cp-16-1581-2020, 2020.

Ciais, P. and Jouzel, J.: Deuterium and oxygen 18 in precipitation: Isotopic model, including mixed cloud processes, J. Geophys. Res. Atmos., 99(D8), 16793–16803, doi:10.1029/94JD00412, 1994.

Ekaykin, A. A., Lipenkov, V. Y., Kuzmina, I. N., Petit, J. R., Masson-Delmotte, V. and Johnsen, S. J.: The changes in isotope composition and accumulation of snow at Vostok station, East Antarctica, over the past 200 years, Ann. Glaciol., 39(1), 569–575, doi:10.3189/172756404781814348, 2004.

Fisher, D. A., Reeh, N. and Clausen, H. B.: Stratigraphic noise in time series derived from ice cores, Ann. Glaciol, 7, 76–83, 1985.

Higbie, J.: Uncertainty in the linear regression slope, Am. J. Phys., 59(2), 184–185, doi:10.1119/1.16607, 1991.

Jouzel, J. and Merlivat, L.: Deuterium and oxygen 18 in precipitation: Modeling of the isotopic effects during snow formation, J. Geophys. Res. Atmos., 89(D7), 11749–11757, doi:10.1029/JD089iD07p11749, 1984.

Laepple, T., Münch, T., Casado, M., Hoerhold, M., Landais, A. and Kipfstuhl, S.: On the similarity and apparent cycles of isotopic variations in East Antarctic snow pits, Cryosph., 12(1), 169–187, doi:10.5194/tc-12-169-2018, 2018.

Münch, T., Kipfstuhl, S., Freitag, J., Meyer, H. and Laepple, T.: Regional climate signal vs. local noise: a two-dimensional view of water isotopes in Antarctic firn at Kohnen Station, Dronning Maud Land, Clim. Past, 12(7), 1565–1581, doi:10.5194/cp-12-1565-2016, 2016.

Münch, T., Werner, M. and Laepple, T.: How precipitation intermittency sets an optimal sampling distance for temperature reconstructions from Antarctic ice cores, Clim. Past, 17(4), 1587–1605, doi:10.5194/cp-17-1587-2021, 2021.

Persson, A., Langen, P. L., Ditlevsen, P. and Vinther, B. M.: The influence of precipitation weighting on interannual variability of stable water isotopes in Greenland, J. Geophys. Res. Atmos., 116(D20), 2011.

Petit, R., Jouzel, J., Pourchet, M. and Merlivat, L.: A detailed study of snow accumulation and stable isotope content in Dome C (Antarctica), J. Geophys. Res. Ocean., 87(C6), 4301–4308, doi:10.1029/JC087iC06p04301, 1982.

Rayleigh, Lord: LIX. On the distillation of binary mixtures, Philos. Mag. Ser. 6, 4(23), 521–537, doi:10.1080/14786440209462876, 1902.

Schoenemann, S. W., Steig, E. J., Ding, Q., Markle, B. R. and Schauer, A. J.: Triple water-isotopologue record from WAIS Divide, Antarctica: Controls on glacial-interglacial changes in 17Oexcess of precipitation, J. Geophys. Res. Atmos., 119(14), 8741–8763, doi:10.1002/2014JD021770, 2014.

Sime, L. C., Marshall, G. J., Mulvaney, R. and Thomas, E. R.: Interpreting temperature information from ice cores along the Antarctic Peninsula: ERA40 analysis, Geophys. Res. Lett., 36(18), 2009.

---

## Author Comment (AC1)

**Author response:** We thank the reviewer for their constructive review of the manuscript. All the suggestions from the reviewer will be addressed (detailed below). Further to this, and in response to the review from Matthieu Casado, we propose to make several key changes to the manuscript that will improve the final version while still retaining much of the work already presented here:

1. We will streamline the analyses included in sections 3.1-3.5 to ensure that:
   a. A single model output is utilized for all calculations (ERA-5);
   b. We will remove the discussion around condensation temperature and focus on the combined effects of precipitation intermittency and surface temperature;
2. We will expand on section 3.6 (Mount Brown South water isotope record) in order to dig more deeply into the impacts of precipitation intermittency on the isotope record at this site
   a. We will remove a discussion around $d_{ln}$, as this does not enhance the text or help to understand the impacts of precipitation intermittency on the stable isotope record;
   b. We will include a discussion on stratigraphic noise, including estimations of the impact of stratigraphic noise at this location based on the two water isotope records available (MBS-Main and MBS-Charlie);
   c. We will better interrogate the relationship between precipitation intermittency and $\delta^{18}O$ using the methods suggested by the reviewer (compare the temperature record to the temperature record with precipitation intermittency, and also with precipitation intermittency but excluding the EPE events);
   d. If possible, we will support the results with an investigation into modelled results where precipitation intermittency is both included and excluded from modelled isotopic values.

While many of these proposed changes are not required by this reviewer, we feel that these changes will address the primary concerns of the other reviewer and result in a much stronger final study.

**Review of Climatology of the Mount Brown South ice core site in East Antarctica: implications for the interpretation of a water isotope record**

It was a pleasure to read Jackson et al. (egusphere-2022-1171). Jackson et al. provide a comprehensive investigation of extreme conditions (snow accumulation rate and temperature) at Mt Brown South in East Antarctica and associated impact on water isotopes. Understanding the relationships between ice core water isotopes and climate over the re-analysis period is critical to interpret the ice core record from the Mount Brown South site. The age model for the sections of core used in this manuscript have previously been published (Crockart et al., 2021). This manuscript is particularly timely as the community is realising the importance of extremes on climate and thus the implications for paleoclimate records. The manuscript is engaging, well written and follows a logical structure. I recommend publication and hope the minor comments below are helpful in improving the manuscript.

Minor suggestions

- Why were 5-day back trajectories run and not 10 or 14-day back trajectories? Please justify and consider expanding to 10–14-day trajectories.

*Author response: Several studies have investigated uncertainties associated with HySPLIT trajectory modelling, and estimate errors of 15-30% on 5-day trajectories (e.g. Scarchilli et al., 2011). Increasing the trajectory length further increases the associated error. While 5-day trajectories likely do not capture the full*

*range of moisture sources, we chose to use 5-day back-trajectories to balance estimations of moisture source with minimisation of error. As such, we believe that 5-day back-trajectories are still the most appropriate choice for this study, but we will provide a detailed explanation for this reasoning in the text.*

- Confusion over the relationship between blocking and winter EPE: L429-430 states no correlation between winter EPE and blocking while L746 states there is a weaker association between winter EPE and blocking. Which is it? It is interesting that the authors find a greater occurrence of extreme accumulation events during the winter but no/weak correlation between EPE accumulation and atmospheric blocking in winter. Please clarify the winter relationship and further explore the causes of the high occurrence of winter EPE.

**Author response:** *We find that there is a positive correlation between total accumulation and winter blocking, but not EPE accumulation. The wording in L746 is misleading, and should read that there is no association between winter EPE and blocking. We will update this, as well as further investigating the drivers of the winter extreme events.*

- Please include the identification of atmospheric rivers in the methods section. Is MBS located in a region that typically experiences atmospheric rivers?

**Author response:** *We will include this in the methods section, and expand on the discussion around the occurrence of atmospheric rivers in this region of Antarctica.*

- Please discuss the variability of the MBS-C and MBS-main d18O records (Fig. 7a).

**Author response:** *In response to comments from the second reviewer, we intent to expand upon the section discussing d18O records, which will include more detailed discussion around the variability of MBS-C and MBS-Main. See response to the review from Matthieu Casado for more details.*

Specific comments
L45-48 Please add reference.
**Author response:** *This will be added.*

L52 and throughout Consider using the terminology enriched/depleted rather than heavy/light.
**Author response:** *When referring to a single isotopologue (i.e. $H_2^{18}O$) then the terms 'heavy' and 'light' will continue to be used, however we will update the terminology throughout to use 'enriched' and 'depleted' when discussing variations in d18O.*

L64-65 Is this the same definition as EPE in Turner et al. (2019). Please clarify and add reference.
**Author response:** *This is the same definition and we will update the text to reflect this.*

L83, L107 and throughout "Wille et al. (2021)" "Vance et al. (2016)"
**Author response:** *This will be updated throughout.*

L107 Please add location of this core.
**Author response:** *The will be added.*

L112 Note that the age model for the full core is still in development. Please update with a reference if this is now published.
**Author response:** *This manuscript is still in preparation, and will likely not be published prior to submission of revisions to this manuscript.*

L124 "...isotope record."
**Author response:** *The will be updated.*

L153 Delete "Only". This is a substantial amount of work and criterial for the interpretation of the longer record.
**Author response:** *We thank the reviewer for this acknowledgement, and will remove "only" from the text.*

L156-157 Please add the time resolution each sample covers.
**Author response:** *The will be added.*

L177 Please state what seasons these markers are assumed or known to occur in.

L182-184 Move last sentence in paragraph to first sentence in paragraph and then you can briefly state how the cores were dated by Crockart et al. (2021). Please add the dating uncertainty at the base of each core.

*Author response: This sentence will be moved, and we will expand on the text to identify the seasonality of each of the chemical species used for dating (and reasoning behind this). We will still keep the section on dating brief as it is discussed in detail in Crockart et al. (2021).*

L259 Why 5-day back trajectories and not 10 or 14 day back trajectories?

*Author response: See above.*

L290-291 Please check significant figures here and throughout.

*Author response: These will be checked and updated.*

L293 RACMO2 slightly underestimates accumulation rates derived from the MBS core. How does this underestimation compare to other ice cores? e.g. Thomas et al. (2017)

*Author response: We will include a reference to this, as well as a discussion around how this compares to other sites from Thomas et al. (2017).*

Thomas, E. R., van Wessem, J. M., Roberts, J., Isaksson, E., Schlosser, E., Fudge, T. J., Vallelonga, P., Medley, B., Lenaerts, J., Bertler, N., van den Broeke, M. R., Dixon, D. A., Frezzotti, M., Stenni, B., Curran, M., and Ekaykin, A. A.: Regional Antarctic snow accumulation over the past 1000 years, Clim. Past, 13, 1491–1513, https://doi.org/10.5194/cp-13-1491-2017, 2017.

Scarchilli, C., M. Frezzotti, and P. M. Ruti, 2011: Snow precipitation at four ice core sites in East Antarctica: Provenance, seasonality and blocking factors. *Climate Dyn.*, **37**, 2107–2125, doi:10.1007/s00382-010-0946-4.

Figure 1 Please add insert to map showing the location of the cores.

---

## Author Comment (AC2)

**Author Response:** We thank Matthieu for a thorough and thoughtful review of the manuscript. In response to the suggestions from the reviewer, we propose several key changes to the manuscript that will improve the final version while still retaining much of the work already presented here:

1. We will streamline the analyses included in sections 3.1-3.5 to ensure that:
   a. A single model output is utilized for all calculations (ERA-5);
   b. We will remove the discussion around condensation temperature and focus on the combined effects of precipitation intermittency and surface temperature;
2. We will expand on section 3.6 (Mount Brown South water isotope record) in order to dig more deeply into the impacts of precipitation intermittency on the isotope record at this site
   a. We will remove a discussion around $d_{ln}$, as this does not enhance the text or help to understand the impacts of precipitation intermittency on the stable isotope record;
   b. We will include a discussion on stratigraphic noise, including estimations of the impact of stratigraphic noise at this location based on the two water isotope records available (MBS-Main and MBS-Charlie);
   c. We will better interrogate the relationship between precipitation intermittency and $\delta^{18}O$ using the methods suggested by the reviewer (compare the temperature record to the temperature record with precipitation intermittency, and also with precipitation intermittency but excluding the EPE events);
   d. If possible, we will support the results with an investigation into modelled results where precipitation intermittency is both included and excluded from modelled isotopic values.

We feel that these changes will address the reviewers primary concern (that the manuscript illustrates the impacts of precipitation intermittency without fully understanding it) and result in a much stronger final study.

**Review of Climatology of the Mount Brown South ice core site in East Antarctica: implications for the interpretation of a water isotope record**

The manuscript describes the climatic conditions at the Mount Brown South ice core drilling site in Antarctica, and the implications for the interpretation of the isotopic composition as a paleoclimatic record. The manuscript combines observations, reanalysis, and back-trajectories analyses to evaluate the contribution of warm synoptic events to the signal. Specifically, the authors address the global contribution of extreme precipitation events, a subset of events which accounts for the largest total precipitation, which includes atmospheric rivers. The results are compared to the upper part of the main ice core from Mount Brown South, as well as from a short replicate core.

Overall, such studies evaluating the impact of the local climatic conditions to the isotopic signal stored in the ice core should be necessary preamble approaches to the interpretation of the isotopic paleothermometer. The authors combine a large range of polar meteorology tools to constrain the impact of precipitation intermittency and seasonality which are two limiting factors to the interpretation of the water isotopes at high resolution. The methodology is appropriate; the figures convey the necessary information. Overall, the manuscript is well written but I believe the following points should be taken into account before it is considered for publication.

**General comments:**
I acknowledge that this is already a very dense manuscript, which includes results ranging across disciplines and provide an in-depth study of the impact of the signal of the largest precipitation events imprinted in the ice core. In my opinion, the manuscript presently falls just short of achieving its goal, i.e. "understand how precipitation intermittency impacts the temperature records preserved in an East Antarctic ice core" (first sentence of the conclusion), because the pertinence of the results for signal analysis is not sufficiently discussed. To move

beyond *illustrating* the impacts of precipitation intermittency on the signal, and really *understand* them, I would suggest including the following discussion elements:

- While it's true that precipitation intermittency, and in particular the fact that synoptic events are associated with warmer than conditions (and thus a warm offset), for the interpretation of the isotopic records, which are always given as an anomaly, it does not matter. The most important question is for each time scale, if the amplitude of the variations of isotopic composition and of temperature can be scaled by the same factor. Section 3.6.2 which evaluates the correlation between isotopic composition and temperature is disconnected from the rest of the studies which address the EPE influence. At this point, new calculations evaluating the modelled isotopic signal with and without the contribution of the EPEs might be out of the framework of the manuscript, but discussing the influence of these large events, building up on the results from (Sime et al., 2009), (Casado et al., 2020), and (Münch et al., 2021) would support the main goal of the manuscript,

*Author response: While in principle we agree that a bias does not matter to an anomaly measurement, we find that the magnitude of this bias is dependent on the proportion of accumulation derived from EPEs, in which case we find that the warm bias can vary substantially from year-to-year. However, as highlighted by the reviewer, we have not considered the influence of stratigraphic noise, which would act as an additional filter, and reduce the inter-annual variability the magnitude of the warm bias from precipitation intermittency. In the revised manuscript, we will address the impacts of stratigraphic noise, which will result in a re-assessment of the conclusions of section 3.6.2 and build upon the previous results (Sime et al., 2009; Casado et al., 2020; Münch et al., 2021);*

- The impact of stratigraphic noise is completely ignored (Ekaykin et al., 2004; Fisher et al., 1985; Münch et al., 2016; Petit et al., 1982), and since it is similar than precipitation intermittency, it needs to be at least discussed. Specifically, here, the distance between the cores will be key to evaluate if the stratigraphic noise will affect the two cores the same way, or not, both in term of dating uncertainty, and of noise added to the isotopic signal.

*Author response: We acknowledge that stratigraphic noise is an important contributor to the water isotope record which can further mask the underlying climate signal preserved in the ice core record. As we have two closely-situated cores, it is possible to estimate the impact of stratigraphic noise at our site and we will add a section discussing this to the manuscript. We similarly will include an estimation of the signal-to-noise ratio based on the two isotope records we have available, and a discussion around how this impacts the temporal resolution of a climatic signal resolvable within the core.*

- Overall, the use of the condensation temperature here is not based on physics, since the temperature time series are simply compared with the isotopic records. Since the condensation temperature is an artificial temperature retrieved from the model at a variable location, which is solely based on the pressure level, and not on where the model is predicting that the precipitation takes place, it does not bring any added value to you. How are the results of Section 3.6 for precipitation weighted surface temperature? While I agree that in principle, the fractionation takes place way above the surface when the moisture condensates, the signal is actually only weakly affected by the local fractionation coefficient sensitivity to temperature when you actually calculate a Rayleigh distillation, and the link between temperature and isotopic composition is acquired by integrating the whole distillation (Rayleigh, 1902).

*Author response: We agree that the discussion of condensation temperature within the text is not supported by a clear physical mechanism. Instead we propose to re-do calculations for surface temperatures, and present only a discussion on surface temperatures within the text.*

- At the end, section 3.6 is disconnected from the section 3.1 to 3.5, even though, this is the important aspect. How much of the results of section 3.6 can you explain from the EPE? Can you compare the temperature record to the temperature record with precipitation intermittency, and also with precipitation intermittency but excluding the EPE events? What does it teach you for the interpretation of the d18O of the MBS core?

*Author response: We agree that section 3.6 is somewhat disconnected from the previous sections, and we will include the suggested comparisons in order to further interrogate the data in order to better understand the true impacts of precipitation intermittency at this site (rather than simply illustrating them, as highlighted by the reviewer at the beginning).*

**Specific comments:**
Line 59: The following studies should be cited here (Sime et al., 2009), (Casado et al., 2020), and (Münch et al., 2021).
*Author response: We will include a discussion and citation of the listed studies here.*

Lines 64-65, and then later on, 83-84: the percentiles at which the EPE are defined should be consistent throughout the manuscript. Either, it needs to be based on other studies (such as the Wille et al, 2021), and then defined just once in the introduction, or, if you chose to apply your own threshold, it needs to be defined in the methods, sensitivity tests need to be applied (more cumbersome).
*Author response: We will use a clear and consistent definition of EPEs through the study (which we will update as the 95th percentile to retain consistency with previous studies).*

Lines 162 – 164: Is 17O-excess actually used in this paper? If it isn't this is unnecessary information to add here.
*Author response: This will be removed from the manuscript as 17O-excess is not discussed further.*

Section 2.2: It is not clear to me why the EPEs are characterised with RACMO and the temperatures, geopotential,… are extracted from ERA5. The dataset (RACMO, ERA5, or other) should here be only a tool used to apply the method (such as the routine from Turner et al, 2019 to identify EPEs), and ideally, it should be reproducible with both tools. Using both datasets as a gridded meteorological data input is not necessarily bad, if both are used completely for all diagnoses, and if both datasets are used for sensitivity tests, but here, it raises the question whether the results can be reproduced using one single dataset only. I believe that a coherent ensemble is more important than "direct comparisons with previous work".
*Author response: We agree that the use of combined datasets is an ineffective, and potentially misleading, way to do these analyses. We will re-do the analyses using a single gridded product (ERA-5)*

Lines 259 to 261: "Here, we use HySPLIT to generate 5-day back-trajectories (120 hours), originating at the MBS site at a height of 1500 m above ground level, which is equivalent to approximately 3500 m above sea level."
Here, you evaluate the formation of precipitation at 1500 m. agl. Later on, the condensation temperature is defined as the 650mbar pressure level (line 554). Is this coherent?
Line 317 (and several times later on): if the error bar is larger than one, the '.2' is not a significant figure.
*Author response: Thank you, we will update errors to reflect this throughout.*

Line 549 to 555: "In Antarctica, a strong inversion layer is present over much of the ice sheet meaning that surface temperature and condensation temperature (above the inversion layer) are not always directly related (Jouzel and Merlivat, 1984). Water isotope records are directly dependent on condensation temperatures rather than surface air temperature. We therefore calculated the temperature anomaly relative to the 30-day seasonal mean temperature for all EPE days identified by RACMO2.3p2 using both surface (t2m) and condensation temperature (approximated at the 650 hPa pressure level) data from ERA-5 (see section 2.2.2).".
This is not exactly true. The local fractionation coefficient is related to the local condensation temperature (Jouzel and Merlivat, 1984), not the water isotopic composition which integrates all the successive fractionation coefficients throughout the Rayleigh distillation (Ciais and Jouzel, 1994; Rayleigh, 1902; Schoenemann et al., 2014). As a result, the most important factor is the rainout fraction (which follows Clausius Clapeyron's law, and thus, is a logarithmic ratio of the local to source temperature). Overall, the fractionation coefficient variation with temperature is only a second order parameter (Bailey et al., 2019). Taking into account the surface temperature or the condensation temperature is then just a convention at this point.
*Author response: We thank the reviewer for this clarification and discussion. As previously discussed, we agree that the use of condensation temperature throughout this manuscript has been poorly handled and we will instead discuss only the relationship of the isotopic record to surface temperature throughout. While this again oversimplifies the controls on the isotopic composition, we feel (as highlighted by the reviewer) that this will be a clearer way to improve understanding of the isotopic record, and in particular provide a consistent approach to understanding how precipitation intermittency and local temperature influence the isotopic record.*

Section 3.5.1: The warm bias induced by the correlation between strong precipitation events and temperature was already used in (Sime et al., 2009), (Persson et al., 2011), and (Casado et al., 2018). The proposed strategy could help enhanced the discussion. Since the purpose is to apply these results to the interpretation of the isotopic signal, it would make sense to evaluate if this is only going to be a bias without any frequency dependency or if this create a signal that affects particularly the high frequency variability of the signal. A bias does not matter for the interpretation of the paleothermometer because only the anomalies are studied.

*Author response: Given that this study forms the basis of understanding a 1200-year climate record, we agree understanding the frequency-dependence of the bias is an important aspect of this study which has been ignored. We will include a discussion around the frequency-dependence of the bias, with consideration to the results presented in the aforementioned studies.*

Lines 622: The use of deuterium excess here appears superfluous and raises more questions than answers: (i) why is the dexcess introduced here, near the end of the Results and Discussion sections, and not in the introduction or in the methods ? (ii) why is the dexcess introduced at all since it's not used in the manuscript ? (the correlation will temperature will be discussed later on) (iii) the non-linearity of the relationship between d18O and dD will not appear on a single site with small variations of temperature, so why is the manuscript using a non-classical definition of dexcess ? (iv) and, in turn, what is the classical d-excess definition looking like? Both definitions have positive and negative aspects, but here, it is not clear what is the benefit of using this definition. Overall, I would recommend to remove the dexcess completely from the manuscript, and focus more on the link between d18O and accumulation, but if you decide to keep the dexcess, it would be necessary to explain why this specific definition is used here, even though, the classical definition has been shown "be more problematic at sites further inland than MBS", and also (and maybe more importantly), at much larger time scales which the variations of temperature were much larger.

*Author response: We agree that a discussion on d-excess/dln does not provide any additional understanding of the relationship between precipitation intermittency and the water isotope record, and will be removed from the manuscript.*

Lines 640: There is no discussion on the stratigraphic noise, and if this could explain why the two cores have weak correlation.

*Author response: A discussion on stratigraphic noise will be included in the revised manuscript.*

Lines 665: Dating ice cores using the water isotopes has been already proven to lead to high errors when the signal to noise ratio is low (Laepple et al., 2018), which could be discussed here.

*Author response: Dating of the core was done primarily using species with clear seasonality ($nssSO_4^{2-}$, $Na^+$, $SO_4^{2-}/Cl^-$, with the Pinatuba eruption (identified as a peak in $nssSO_4^{2-}$) used as a tie point between records (section 2.1.3), with water isotopes used as a secondary dating tool. However, we realise that this sentence (line 665) suggest that $\delta^{18}O$ was the primary dating tool. We will re-write this to clarify that there is weak cyclicity across all of the species used for dating during this period, including $\delta^{18}O$.*

Lines 671 to 677: What is the conclusion from the piece of information? The unexpected correlation between dexcess and temperature could be link with three possible explanations: (i) the definition of dexcess is not adapted and a lot of the $1_{st}$ order (d18O) signal remains in the dataset, (ii) it is actually a source temperature signal, but condensation and source temperature are correlated and this explains this link, or (iii), this is a complete random correlation (according to the pvalue, there is 0.7% chance of this happening).
Since there is a strong positive correlation between the dln and d18O in figure 7, it seems likely that the use of dexcess here (using either the classical or log definition) is not bringing additional information, but this should be tested if the dexcess is kept in the manuscript.

*Author response: We agree that here (and throughout the manuscript) the inclusion of dln/dxs is superfluous and does not enhance the text – nor is there a physical reason that we might expect to see a relationship between dln and the local temperature. As such, we believe that the text will be enhanced and streamlined by removal of discussions surrounding dln and will focus solely on the relationship between d18O and site temperature.*

Line 696 to 697: The error bars appear very small considering the relatively low coefficient correlation (Higbie, 1991), I would suggest rather 0.2 and 0.3 respectively. A simple test can be to compare the slope of d18O with the temperature to the inverse of the slope of the temperature to d18O (which should be equal normally), but sometimes offer difference which can be used to evaluate the error bars.

*Author response: We will re-asses the error bars associated with the calculated slopes and perform the suggested test to ensure these are presented accurately.*

Lines 696 compared to the previous paragraph: it seems here that you are comparing the condensation temperature to isotope slopes from your results to mostly surface temperature datasets. How is this affecting your results? Does using the condensation temperature really help the study? Overall, the compared data should be coherent, especially because variations of surface temperature are not scaled one to one with condensation temperature.

*Author response: Again, we agree that the use of condensation temperatures here (and throughout the text) is not beneficial to the study and we will re-do calculations to consider surface temperatures instead.*

Bibliography

Bailey, A., Singh, H. K. A. and Nusbaumer, J.: Evaluating a Moist Isentropic Framework for Poleward Moisture Transport: Implications for Water Isotopes Over Antarctica, Geophys. Res. Lett., 46(13), 7819–7827, doi:https://doi.org/10.1029/2019GL082965, 2019.

Casado, M., Landais, A., Picard, G., Münch, T., Laepple, T., Stenni, B., Dreossi, G., Ekaykin, A., Arnaud, L., Genthon, C., Touzeau, A., Masson-Delmotte, V. and Jouzel, J.: Archival processes of the water stable isotope signal in East Antarctic ice cores, Cryosph., 12(5), 1745–1766, doi:10.5194/tc-12-1745-2018, 2018.

Casado, M., Münch, T. and Laepple, T.: Climatic information archived in ice cores: impact of intermittency and diffusion on the recorded isotopic signal in Antarctica, Clim. Past, 16(4), 1581–1598, doi:10.5194/cp-16-1581-2020, 2020.

Ciais, P. and Jouzel, J.: Deuterium and oxygen 18 in precipitation: Isotopic model, including mixed cloud processes, J. Geophys. Res. Atmos., 99(D8), 16793–16803, doi:10.1029/94JD00412, 1994.

Ekaykin, A. A., Lipenkov, V. Y., Kuzmina, I. N., Petit, J. R., Masson-Delmotte, V. and Johnsen, S. J.: The changes in isotope composition and accumulation of snow at Vostok station, East Antarctica, over the past 200 years, Ann. Glaciol., 39(1), 569–575, doi:10.3189/172756404781814348, 2004.

Fisher, D. A., Reeh, N. and Clausen, H. B.: Stratigraphic noise in time series derived from ice cores, Ann. Glaciol, 7, 76–83, 1985.

Higbie, J.: Uncertainty in the linear regression slope, Am. J. Phys., 59(2), 184–185, doi:10.1119/1.16607, 1991.

Jouzel, J. and Merlivat, L.: Deuterium and oxygen 18 in precipitation: Modeling of the isotopic effects during snow formation, J. Geophys. Res. Atmos., 89(D7), 11749–11757, doi:10.1029/JD089iD07p11749, 1984.

Laepple, T., Münch, T., Casado, M., Hoerhold, M., Landais, A. and Kipfstuhl, S.: On the similarity and apparent cycles of isotopic variations in East Antarctic snow pits, Cryosph., 12(1), 169–187, doi:10.5194/tc-12-169-2018, 2018.

Münch, T., Kipfstuhl, S., Freitag, J., Meyer, H. and Laepple, T.: Regional climate signal vs. local noise: a two-dimensional view of water isotopes in Antarctic firn at Kohnen Station, Dronning Maud Land, Clim. Past, 12(7), 1565–1581, doi:10.5194/cp-12-1565-2016, 2016.

Münch, T., Werner, M. and Laepple, T.: How precipitation intermittency sets an optimal sampling distance for temperature reconstructions from Antarctic ice cores, Clim. Past, 17(4), 1587–1605, doi:10.5194/cp-17-1587-2021, 2021.

Persson, A., Langen, P. L., Ditlevsen, P. and Vinther, B. M.: The influence of precipitation weighting on interannual variability of stable water isotopes in Greenland, J. Geophys. Res. Atmos., 116(D20), 2011.

Petit, R., Jouzel, J., Pourchet, M. and Merlivat, L.: A detailed study of snow accumulation and stable isotope content in Dome C (Antarctica), J. Geophys. Res. Ocean., 87(C6), 4301–4308, doi:10.1029/JC087iC06p04301, 1982.

Rayleigh, Lord: LIX. On the distillation of binary mixtures, Philos. Mag. Ser. 6, 4(23), 521–537, doi:10.1080/14786440209462876, 1902.

Schoenemann, S. W., Steig, E. J., Ding, Q., Markle, B. R. and Schauer, A. J.: Triple water-isotopologue record from WAIS Divide, Antarctica: Controls on glacial-interglacial changes in 17Oexcess of precipitation, J. Geophys. Res. Atmos., 119(14), 8741–8763, doi:10.1002/2014JD021770, 2014.

Sime, L. C., Marshall, G. J., Mulvaney, R. and Thomas, E. R.: Interpreting temperature information from ice cores along the Antarctic Peninsula: ERA40 analysis, Geophys. Res. Lett., 36(18), 2009.

---

## Author Response (AR1)

**Author Response:** We thank both reviewers for their thoughtful comments on the manuscript. In response to suggestions from both reviewers, we have made several major improvements to the original manuscript. These major changes are highlighted here: We will streamline the analyses included in sections 3.1-3.5 to ensure that:

       a. A single model output is utilized for all calculations (ERA-5);

       b. T discussion around condensation temperature has been removed and instead the manuscript focuses on the impact of extreme events on surface temperature and the water isotope record;

  2. Section 3.6 (Mount Brown South water isotope record) has been largely re-written and modified in order to better address the question of the impact of extreme events on the ice core water isotope recordd.

       a. The discussion around $d_{ln}$ has been removed, as this does not enhance the text or help to understand the impacts of precipitation intermittency on the stable isotope record;

       b. We have included a discussion on stratigraphic noise;

       c. We have better interrogated the relationship between precipitation intermittency and $\delta^{18}O$ using the methods suggested by the reviewer (compare the temperature record to the temperature record with precipitation intermittency, and also with precipitation intermittency but excluding the EPE events);

We feel that these changes address the reviewers primary concerns and result in a much stronger final study.

**Review of Climatology of the Mount Brown South ice core site in East Antarctica: implications for the interpretation of a water isotope record**

The manuscript describes the climatic conditions at the Mount Brown South ice core drilling site in Antarctica, and the implications for the interpretation of the isotopic composition as a paleoclimatic record. The manuscript combines observations, reanalysis, and back-trajectories analyses to evaluate the contribution of warm synoptic events to the signal. Specifically, the authors address the global contribution of extreme precipitation events, a subset of events which accounts for the largest total precipitation, which includes atmospheric rivers. The results are compared to the upper part of the main ice core from Mount Brown South, as well as from a short replicate core.

Overall, such studies evaluating the impact of the local climatic conditions to the isotopic signal stored in the ice core should be necessary preamble approaches to the interpretation of the isotopic paleothermometer. The authors combine a large range of polar meteorology tools to constrain the impact of precipitation intermittency and seasonality which are two limiting factors to the interpretation of the water isotopes at high resolution. The methodology is appropriate; the figures convey the necessary information. Overall, the manuscript is well written but I believe the following points should be taken into account before it is considered for publication.

**General comments:**
I acknowledge that this is already a very dense manuscript, which includes results ranging across disciplines and provide an in-depth study of the impact of the signal of the largest precipitation events imprinted in the ice core. In my opinion, the manuscript presently falls just short of achieving its goal, i.e. "understand how precipitation intermittency impacts the temperature records preserved in an East Antarctic ice core" (first sentence of the conclusion), because the pertinence of the results for signal analysis is not sufficiently discussed. To move beyond *illustrating* the impacts of precipitation intermittency on the signal, and really *understand* them, I would suggest including the following discussion elements:

- While it's true that precipitation intermittency, and in particular the fact that synoptic events are associated with warmer than conditions (and thus a warm offset), for the interpretation of the isotopic records, which are always given as an anomaly, it does not matter. The most important question is for each time scale, if the amplitude of the variations of isotopic composition and of temperature can be scaled by the same factor. Section

3.6.2 which evaluates the correlation between isotopic composition and temperature is disconnected from the rest of the studies which address the EPE influence. At this point, new calculations evaluating the modelled isotopic signal with and without the contribution of the EPEs might be out of the framework of the manuscript, but discussing the influence of these large events, building up on the results from (Sime et al., 2009), (Casado et al., 2020), and (Münch et al., 2021) would support the main goal of the manuscript,

**Author response:** We thank the reviewer for this useful comment. We have updated both sections 3.5.1 (Temperature Bias) and section 3.6 (Mount Brown South water isotope record) to better explore the EPE influence. In particular, we have now included a discussion about the temperature bias for a precipitation-weighted temperature record both with and without EPEs, and provided a more detailed discussion around how it is the variability of the temperature bias that impacts the time-scale for which a climate signal can be resolved.

- The impact of stratigraphic noise is completely ignored (Ekaykin et al., 2004; Fisher et al., 1985; Münch et al., 2016; Petit et al., 1982), and since it is similar than precipitation intermittency, it needs to be at least discussed. Specifically, here, the distance between the cores will be key to evaluate if the stratigraphic noise will affect the two cores the same way, or not, both in term of dating uncertainty, and of noise added to the isotopic signal.

**Author response:** We have added in a section on stratigraphic noise in the text (section 3.6.4). In particular, we discuss how stratigraphic noise impacts each core independently due to the distance of the cores from one another (94 m), resulting in a decorrelation between the two records independent from the noise that results from EPEs.

- Overall, the use of the condensation temperature here is not based on physics, since the temperature time series are simply compared with the isotopic records. Since the condensation temperature is an artificial temperature retrieved from the model at a variable location, which is solely based on the pressure level, and not on where the model is predicting that the precipitation takes place, it does not bring any added value to you. How are the results of Section 3.6 for precipitation weighted surface temperature? While I agree that in principle, the fractionation takes place way above the surface when the moisture condensates, the signal is actually only weakly affected by the local fractionation coefficient sensitivity to temperature when you actually calculate a Rayleigh distillation, and the link between temperature and isotopic composition is acquired by integrating the whole distillation (Rayleigh, 1902).

**Author response:** We have removed discussions around condensation temperature and instead focused on surface temperature throughout. In particular, section 3.5.1 (Temperature bias) has been re-written and calculations re-done using surface temperature data instead.

- At the end, section 3.6 is disconnected from the section 3.1 to 3.5, even though, this is the important aspect. How much of the results of section 3.6 can you explain from the EPE? Can you compare the temperature record to the temperature record with precipitation intermittency, and also with precipitation intermittency but excluding the EPE events? What does it teach you for the interpretation of the d18O of the MBS core?

**Author response:** We thank the reviewer for this comment, as we feel that addressing this more clearly has improved the manuscript. We have better integrated section 3.6 with the rest of the manuscript in several ways:
1. Comparison of the relationship between d18O and mean annual temperature, precipitation-weighted mean annual temperature, precipitation-weighted mean annual temperature without EPEs, and precipitation-weighted temperature with only EPEs. This allows us to better explore the relationship between water isotopes and extreme events at this site that was previously poorly explored.
2. More detailed discussion tying the water isotope record to the previous sections (transport pathways, synoptic conditions etc).

We feel that these changes help to integrate the section on the water isotope records of the MBS ice cores with the rest of the manuscript, rather than including it as a disconnected section.

**Specific comments:**
Line 59: The following studies should be cited here (Sime et al., 2009), (Casado et al., 2020), and (Münch et al., 2021).
**Author response:** This has been updated:
"This can result in potential signal bias where climate during certain synoptic conditions is preserved and recorded in the accumulated snowfall which may not necessarily be representative of the mean climatology (Turner et al., 2019; Sime et al., 2009; Casado et al., 2020; Münch et al., 2021)."

Lines 64-65, and then later on, 83-84: the percentiles at which the EPE are defined should be consistent throughout the manuscript. Either, it needs to be based on other studies (such as the Wille et al, 2021), and then defined just once in the introduction, or, if you chose to apply your own threshold, it needs to be defined in the methods, sensitivity tests need to be applied (more cumbersome).
**Author response:** We have updated the definition of EPEs to be the 90th percentile throughout (as defined in Turner et al. (2019). This is also clearly identified in the introduction:
"We refer to the infrequent large events as extreme precipitation events, or EPEs and define these as representing days where daily snowfall amount is in the 90th percentile or higher, which is consistent with the previous definition from Turner et al. (2019)."

Lines 162 – 164: Is $_{17}$O-excess actually used in this paper? If it isn't this is unnecessary information to add here.
**Author response:** We have updated the methods to remove reference to 17O-excess.

Section 2.2: It is not clear to me why the EPEs are characterised with RACMO and the temperatures, geopotential,… are extracted from ERA5. The dataset (RACMO, ERA5, or other) should here be only a tool used to apply the method (such as the routine from Turner et al, 2019 to identify EPEs), and ideally, it should be reproducible with both tools. Using both datasets as a gridded meteorological data input is not necessarily bad, if both are used completely for all diagnoses, and if both datasets are used for sensitivity tests, but here, it raises the question whether the results can be reproduced using one single dataset only. I believe that a coherent ensemble is more important than "direct comparisons with previous work".
**Author response:** The manuscript has been updated to use ERA-5 data throughout to maintain consistency in the datasets.

Lines 259 to 261: "Here, we use HySPLIT to generate 5-day back-trajectories (120 hours), originating at the MBS site at a height of 1500 m above ground level, which is equivalent to approximately 3500 m above sea level."
Here, you evaluate the formation of precipitation at 1500 m. agl. Later on, the condensation temperature is defined as the 650mbar pressure level (line 554). Is this coherent?
**Author response:** We have removed the discussion around condensation temperature.

Line 317 (and several times later on): if the error bar is larger than one, the '.2' is not a significant figure.
**Author response:** This has been updated throughout.

Line 549 to 555: "In Antarctica, a strong inversion layer is present over much of the ice sheet meaning that surface temperature and condensation temperature (above the inversion layer) are not always directly related (Jouzel and Merlivat, 1984). Water isotope records are directly dependent on condensation temperatures rather than surface air temperature. We therefore calculated the temperature anomaly relative to the 30-day seasonal mean temperature for all EPE days identified by RACMO2.3p2 using both surface (t2m) and condensation temperature (approximated at the 650 hPa pressure level) data from ERA-5 (see section 2.2.2).".
This is not exactly true. The local fractionation coefficient is related to the local condensation temperature (Jouzel and Merlivat, 1984), not the water isotopic composition which integrates all the successive fractionation coefficients throughout the Rayleigh distillation (Ciais and Jouzel, 1994; Rayleigh, 1902; Schoenemann et al., 2014). As a result, the most important factor is the rainout fraction (which follows Clausius Clapeyron's law, and thus, is a logarithmic ratio of the local to source temperature). Overall, the fractionation coefficient variation with temperature is only a second order parameter (Bailey et al., 2019). Taking into account the surface temperature or the condensation temperature is then just a convention at this point.
**Author response:** We thank the reviewer for this clarification and discussion. We agree that the use of condensation temperature throughout this manuscript has been poorly handled and have simplified he manuscript to only consider surface temperature throughout.

Section 3.5.1: The warm bias induced by the correlation between strong precipitation events and temperature was already used in (Sime et al., 2009), (Persson et al., 2011), and (Casado et al., 2018). The proposed strategy could help enhanced the discussion. Since the purpose is to apply these results to the interpretation of the isotopic signal, it would make sense to evaluate if this is only going to be a bias without any frequency dependency or if this create a signal that affects particularly the high frequency variability of the signal. A bias does not matter for the interpretation of the paleothermometer because only the anomalies are studied.
**Author response:** Section 3.5.1 has been updated to include the references above. We agree that further investigations into the frequency dependency of the bias is valid and would improve the manuscript, however,

due to the density of the manuscript already this has not been included. We have instead more clearly highlighted that it is the variability of the bias, rather than the bias itself, that reduces the ability of the isotopes to record inter-annual temperature variability (sections 3.5.1 and 3.6.2).

Lines 622: The use of deuterium excess here appears superfluous and raises more questions than answers: (i) why is the dexcess introduced here, near the end of the Results and Discussion sections, and not in the introduction or in the methods ? (ii) why is the dexcess introduced at all since it's not used in the manuscript ? (the correlation will temperature will be discussed later on) (iii) the non-linearity of the relationship between d18O and dD will not appear on a single site with small variations of temperature, so why is the manuscript using a non-classical definition of dexcess ? (iv) and, in turn, what is the classical d-excess definition looking like? Both definitions have positive and negative aspects, but here, it is not clear what is the benefit of using this definition. Overall, I would recommend to remove the dexcess completely from the manuscript, and focus more on the link between d18O and accumulation, but if you decide to keep the dexcess, it would be necessary to explain why this specific definition is used here, even though, the classical definition has been shown "be more problematic at sites further inland than MBS", and also (and maybe more importantly), at much larger time scales which the variations of temperature were much larger.
**Author response:** We have removed discussion about dexcess from the manuscript, and thank the reviewer for these comments.

Lines 640: There is no discussion on the stratigraphic noise, and if this could explain why the two cores have weak correlation.
**Author response:** A discussion on stratigraphic noise has now been included (Section 3.6.4).

Lines 665: Dating ice cores using the water isotopes has been already proven to lead to high errors when the signal to noise ratio is low (Laepple et al., 2018), which could be discussed here.
**Author response:** Dating of the core was done primarily using species with clear seasonality ($nssSO_4^{2-}$, $Na^+$, $SO_4^{2-}/Cl^-$, with the Pinatuba eruption (identified as a peak in $nssSO_4^{2-}$) used as a tie point between records (section 2.1.3), with water isotopes used as a secondary dating tool. This has been more clearly identified in the manuscript (section 2.1.3) and line 665 has been removed during the revisions on section 3.6.

Lines 671 to 677: What is the conclusion from the piece of information? The unexpected correlation between dexcess and temperature could be link with three possible explanations: (i) the definition of dexcess is not adapted and a lot of the $1_{st}$ order (d18O) signal remains in the dataset, (ii) it is actually a source temperature signal, but condensation and source temperature are correlated and this explains this link, or (iii), this is a complete random correlation (according to the pvalue, there is 0.7% chance of this happening).
Since there is a strong positive correlation between the dln and d18O in figure 7, it seems likely that the use of dexcess here (using either the classical or log definition) is not bringing additional information, but this should be tested if the dexcess is kept in the manuscript.
**Author response:** The discussion of dxs has been removed from the text.

Line 696 to 697: The error bars appear very small considering the relatively low coefficient correlation (Higbie, 1991), I would suggest rather 0.2 and 0.3 respectively. A simple test can be to compare the slope of d18O with the temperature to the inverse of the slope of the temperature to d18O (which should be equal normally), but sometimes offer difference which can be used to evaluate the error bars.
**Author response:** This section of the original manuscript has been removed during the revisions to section 3.6.

Lines 696 compared to the previous paragraph: it seems here that you are comparing the condensation temperature to isotope slopes from your results to mostly surface temperature datasets. How is this affecting your results? Does using the condensation temperature really help the study? Overall, the compared data should be coherent, especially because variations of surface temperature are not scaled one to one with condensation temperature.
**Author response:** The manuscript has been updated to only include references to surface temperature throughout, and comparisons between the water isotopes and temperature records now only consider surface temperature.

Bibliography

Bailey, A., Singh, H. K. A. and Nusbaumer, J.: Evaluating a Moist Isentropic Framework for Poleward Moisture Transport: Implications for Water Isotopes Over Antarctica, Geophys. Res. Lett., 46(13), 7819–7827, doi:https://doi.org/10.1029/2019GL082965, 2019.

Casado, M., Landais, A., Picard, G., Münch, T., Laepple, T., Stenni, B., Dreossi, G., Ekaykin, A., Arnaud, L., Genthon, C., Touzeau, A., Masson-Delmotte, V. and Jouzel, J.: Archival processes of the water stable isotope signal in East Antarctic ice cores, Cryosph., 12(5), 1745–1766, doi:10.5194/tc-12-1745-2018, 2018.

Casado, M., Münch, T. and Laepple, T.: Climatic information archived in ice cores: impact of intermittency and diffusion on the recorded isotopic signal in Antarctica, Clim. Past, 16(4), 1581–1598, doi:10.5194/cp-16-1581-2020, 2020.

Ciais, P. and Jouzel, J.: Deuterium and oxygen 18 in precipitation: Isotopic model, including mixed cloud processes, J. Geophys. Res. Atmos., 99(D8), 16793–16803, doi:10.1029/94JD00412, 1994.

Ekaykin, A. A., Lipenkov, V. Y., Kuzmina, I. N., Petit, J. R., Masson-Delmotte, V. and Johnsen, S. J.: The changes in isotope composition and accumulation of snow at Vostok station, East Antarctica, over the past 200 years, Ann. Glaciol., 39(1), 569–575, doi:10.3189/172756404781814348, 2004.

Fisher, D. A., Reeh, N. and Clausen, H. B.: Stratigraphic noise in time series derived from ice cores, Ann. Glaciol, 7, 76–83, 1985.

Higbie, J.: Uncertainty in the linear regression slope, Am. J. Phys., 59(2), 184–185, doi:10.1119/1.16607, 1991.

Jouzel, J. and Merlivat, L.: Deuterium and oxygen 18 in precipitation: Modeling of the isotopic effects during snow formation, J. Geophys. Res. Atmos., 89(D7), 11749–11757, doi:10.1029/JD089iD07p11749, 1984.

Laepple, T., Münch, T., Casado, M., Hoerhold, M., Landais, A. and Kipfstuhl, S.: On the similarity and apparent cycles of isotopic variations in East Antarctic snow pits, Cryosph., 12(1), 169–187, doi:10.5194/tc-12-169-2018, 2018.

Münch, T., Kipfstuhl, S., Freitag, J., Meyer, H. and Laepple, T.: Regional climate signal vs. local noise: a two-dimensional view of water isotopes in Antarctic firn at Kohnen Station, Dronning Maud Land, Clim. Past, 12(7), 1565–1581, doi:10.5194/cp-12-1565-2016, 2016.

Münch, T., Werner, M. and Laepple, T.: How precipitation intermittency sets an optimal sampling distance for temperature reconstructions from Antarctic ice cores, Clim. Past, 17(4), 1587–1605, doi:10.5194/cp-17-1587-2021, 2021.

Persson, A., Langen, P. L., Ditlevsen, P. and Vinther, B. M.: The influence of precipitation weighting on interannual variability of stable water isotopes in Greenland, J. Geophys. Res. Atmos., 116(D20), 2011.

Petit, R., Jouzel, J., Pourchet, M. and Merlivat, L.: A detailed study of snow accumulation and stable isotope content in Dome C (Antarctica), J. Geophys. Res. Ocean., 87(C6), 4301–4308, doi:10.1029/JC087iC06p04301, 1982.

Rayleigh, Lord: LIX. On the distillation of binary mixtures, Philos. Mag. Ser. 6, 4(23), 521–537, doi:10.1080/14786440209462876, 1902.

Schoenemann, S. W., Steig, E. J., Ding, Q., Markle, B. R. and Schauer, A. J.: Triple water-isotopologue record from WAIS Divide, Antarctica: Controls on glacial-interglacial changes in 17Oexcess of precipitation, J. Geophys. Res. Atmos., 119(14), 8741–8763, doi:10.1002/2014JD021770, 2014.

Sime, L. C., Marshall, G. J., Mulvaney, R. and Thomas, E. R.: Interpreting temperature information from ice cores along the Antarctic Peninsula: ERA40 analysis, Geophys. Res. Lett., 36(18), 2009.

**Review of Climatology of the Mount Brown South ice core site in East Antarctica: implications for the interpretation of a water isotope record**

It was a pleasure to read Jackson et al. (egusphere-2022-1171). Jackson et al. provide a comprehensive investigation of extreme conditions (snow accumulation rate and temperature) at Mt Brown South in East Antarctica and associated impact on water isotopes. Understanding the relationships between ice core water isotopes and climate over the re-analysis period is critical to interpret the ice core record from the Mount Brown South site. The age model for the sections of core used in this manuscript have previously been published (Crockart et al., 2021). This manuscript is particularly timely as the community is realising the importance of extremes on climate and thus the implications for paleoclimate records. The manuscript is engaging, well written and follows a logical structure. I recommend publication and hope the minor comments below are helpful in improving the manuscript.

Minor suggestions

- •Why were 5-day back trajectories run and not 10 or 14-day back trajectories? Please justify and consider expanding to 10–14-day trajectories.

*Author response: We have updated this section to include reasoning for why 5 day back trajectories have been used rather than 10-14 day trajectories.*
*"A previous investigation into the uncertainties associated with HySPLIT trajectory modelling have estimated errors of 15-20 % on 5-day back-trajectories (Scarchilli et al., 2011). The 5-day back-trajectories used in this study likely do not capture the full range of moisture sources. However, increasing the trajectory length leads to increases in the error associated with the calculations. 5-day back trajectories are used here to provide a balance between estimating moisture sources and transportation pathways while minimising the error associated with these calculations."*

- Confusion over the relationship between blocking and winter EPE: L429-430 states no correlation between winter EPE and blocking while L746 states there is a weaker association between winter EPE and blocking. Which is it? It is interesting that the authors find a greater occurrence of extreme accumulation events during the winter but no/weak correlation between EPE accumulation and atmospheric blocking in winter. Please clarify the winter relationship and further explore the causes of the high occurrence of winter EPE.

*Author response: The use of ERA-5 data both for calculating the blocking index and for identification of EPEs has resulted in slightly different results for this section. Section 3.2.3. has been updated to reflect this. This section now reads:*
*"For all seasons except the austral summer (DJF), there is no correlation between smaller (non-EPE) accumulation and blocking. This indicates that small-scale precipitation events at MBS can occur regardless of atmospheric blocking and are likely due to a variety of transport mechanisms. However, during the summer months when there is reduced storminess in the Southern Ocean (Nakamura and Shimpo, 2004; Trenberth, 1991), blocking still provides an important mechanism for driving even smaller scale precipitation.*

*There is a strong positive correlation between EPE accumulation and atmospheric blocking across all seasons except for the austral spring (SON), indicating that large precipitative events throughout most of the year occur in association with atmospheric blocking in the mid-latitudes. Similarly, blocking is associated with total accumulation during the austral summer (DJF) and winter (JJA), but not during the spring and autumn (MAM and SON).*

*This suggests that atmospheric blocking is an important mechanism for driving precipitation at MBS throughout most of the year, but particularly during the summer (DJF) and winter (JJA) months. Extreme precipitation is particularly dependent on blocking conditions in the Southern Indian Ocean, as indicated by the strong positive correlation between the blocking index and EPE accumulation during all seasons except for SON. A strong positive geopotential height anomaly is observed for all seasons during EPEs (Fig. A3 in the Appendix)."*

- Please include the identification of atmospheric rivers in the methods section. Is MBS located in a region that typically experiences atmospheric rivers?

*Author response: Identification of ARs has now been included in the methods section, and we have included a sentence highlighting that this region is particularly susceptible to ARs.*

"The Amery Ice Shelf region, located nearby and to the west of the MBS ice core site, was identified as having particularly high accumulation from ARs in the Wille et al. (2021) study (~20%), compared with 10-20% across much of the rest of East Antarctica."

- Please discuss the variability of the MBS-C and MBS-main d18O records (Fig. 7a).

*Author response: Section 3.6: Mount Brown South water isotope record has been expanded. In particular, we include a section (3.6.1: The MBS d18O records) which now discusses the variability of the MBS-Charlie and MBS-Main ice core records.*

*"The MBS-Main and MBS-Charlie $\delta^{18}O$ both show strong seasonal cycles, with enriched values during the summer months and depleted values during the winter months (Fig. 7). The amplitudes of the cycles vary greatly from year to year, with the largest amplitude seasonal cycles of ~ 10 ‰ and the smallest amplitudes of ~ 1 ‰. Clear seasonality is observed in both cores during certain periods (i.e. 1990-1993) and much weaker seasonality observed in other years (i.e. 1994-1999). For both records, the most enriched $\delta^{18}O$ values are observed in the summer of 1989/1990 ($\delta^{18}O \approx -25$ ‰). The coincidence of this prominent peak in both records gives confidence in the dating process and the alignment of the two records.*
*"*

**Specific comments**

L45-48 Please add reference.
*Author response: Reference has been added:*
*"Coastal or near-coastal ice cores located in regions of high-accumulation are increasingly being used to help frame modern anthropogenic warming in the context of natural climate variability in Antarctica, as these can provide annually-resolved records capable of recording interannual to decadal fluctuations in the climate (e.g. Thomas et al., 2009; Jones et al., 2016; Stenni et al., 2017)."*

L52 and throughout Consider using the terminology enriched/depleted rather than heavy/light.
*Author response: When referring to a single isotopologue (i.e. $H_2^{18}O$) then the terms 'heavy' and 'light' will continue to be used, however we will update the terminology throughout to use 'enriched' and 'depleted' when discussing variations in d18O.*

L64-65 Is this the same definition as EPE in Turner et al. (2019). Please clarify and add reference.
*Author response: We have clarified this in the text such that it now reads:*
*"We refer to the infrequent large events as extreme precipitation events, or EPEs and define these as representing days where daily snowfall amount is in the 90th percentile or higher, which is consistent with the previous definition from Turner et al. (2019)."*

L83, L107 and throughout "Wille et al. (2021)" "Vance et al. (2016)"
*Author response: This has been updated throughout.*

L107 Please add location of this core.
*Author response: This has been updated.*
*"An extensive site-selection study by Vance et al. (2016) identified Mount Brown South (69.11°S 86.31°E, 2084 m elevation) as a promising location for a new ice core that would likely provide unique climate signals and be complementary to the Law Dome ice core record."*

L112 Note that the age model for the full core is still in development. Please update with a reference if this is now published.
*Author response: This manuscript is still in preparation, and has not be published prior to submission of revisions to this manuscript.*

L124 "...isotope record."
*Author response: This has been updated.*

L153 Delete "Only". This is a substantial amount of work and criterial for the interpretation of the longer record.
*Author response: We thank the reviewer for this acknowledgement, this has been deleted.*

L156-157 Please add the time resolution each sample covers.
*Author response: We have updated this section to include the time resolution of the sampling as follows:*

*"3 cm sampling yields a mean sample resolution of 10 samples year⁻¹. The higher resolution sampling (1.5 cm) yields a mean sample resolution of 20 samples year⁻¹, but due to the effects of diffusion on the stable water isotope record this increased time resolution does not result in enhanced signal preservation."*

L177 Please state what seasons these markers are assumed or known to occur in.

L182-184 Move last sentence in paragraph to first sentence in paragraph and then you can briefly state how the cores were dated by Crockart et al. (2021). Please add the dating uncertainty at the base of each core.

***Author response:*** *Section 2.1.3 has been updated to address the above concerns, and now reads as follows:*

*"Details of the dating procedures and accumulation calculations for the short cores and upper section of MBS-Main can be found in Crockart et al., 2021. Chemical species with clear seasonality (i.e. non sea salt sulphate ($nssSO_4^{2-}$), sodium ($Na^+$), the ratio of sulphate to chloride ($SO_4^{2-}/Cl^-$)) were used in conjunction with water isotope measurements to identify annual layers in MBS-Alpha, MBS-Charlie and the upper portion of MBS-Main. Annual summer horizons were aligned with the sea salt ($Na^+$) minima and maxima in the $nssSO_4^{2-}$ and $SO_4^{2-}/Cl^-$ ratio. Sea salt minima during the summer months reflect the reduced sea-salt aerosol input due to reduced storminess in the Southern Ocean. In contrast, the summer peaks in $nssSO_4^{2-}$ and $SO_4^{2-}/Cl^-$ reflect enhanced biological productivity during the summer months, with the oxidation of biologically produced dimethyl sulphide representing the major $SO_4^{2-}$ source external to sea salts. Layer thicknesses were combined with an empirical density model to determine annual ice equivalent snow accumulation rates.*

*The Pinatubo eruption (mid-1991) was identified as a peak in $nssSO_4^{2-}$ in all cores and was used as a marker to confirm the accuracy of layer counting. Weak seasonality in the years preceding the Pinatubo eruption (1986-1990) introduces an estimated dating uncertainty of ±2 years at the base of the ice core sections considered here (1979)."*

L259 Why 5-day back trajectories and not 10 or 14 day back trajectories?

***Author response****: See above.*

L290-291 Please check significant figures here and throughout.

**Author response:** These have been updated throughout.

L293 RACMO2 slightly underestimates accumulation rates derived from the MBS core. How does this underestimation compare to other ice cores? e.g. Thomas et al. (2017)

***Author response****: Due to changes to the manuscript to utilise output from ERA-5 as opposed to RACMO2.3p2 (see response to reviewer 2), we have added an additional paragraph to discuss the comparisons of ERA-5 with accumulation in Antarctica, which includes the Thomas reference above. This additional paragraph is as follows:*

*"Previous studies have looked at spatial comparisons between observational measurements of accumulation (i.e. from ice cores or snow stakes) and model outputs. Wang et al., 2021 found that ERA-5 captures > 70 % of the surface mass balance observations from Law Dome in East Wilkes Land, and generally captures inter-annual variability across most of the Antarctic ice sheet. Similarly, Tetzner et al. (2019) demonstrated improved performance of ERA-5 at capturing accumulation in the Antarctic Peninsula region compared to ERA-Interim. Comparisons of surface mass estimates from ice core records across Antarctica with outputs from ERA-Interim and RACMO2.3p2 found that regional ice core accumulation composites capture 25-40 % of interannual regional variance (Thomas et al., 2017). In particular, composite ice core surface mass balance records from the Wilkes Land region were shown to be representative of regional surface mass balance derived from both RACMO2.3p2 and ERA-Interim. These previous studies provide further evidence that ERA-5 accurately captures accumulation at this location."*

Thomas, E. R., van Wessem, J. M., Roberts, J., Isaksson, E., Schlosser, E., Fudge, T. J., Vallelonga, P., Medley, B., Lenaerts, J., Bertler, N., van den Broeke, M. R., Dixon, D. A., Frezzotti, M., Stenni, B., Curran, M., and Ekaykin, A. A.: Regional Antarctic snow accumulation over the past 1000 years, Clim. Past, 13, 1491–1513, https://doi.org/10.5194/cp-13-1491-2017, 2017.

Scarchilli, C., M. Frezzotti, and P. M. Ruti, 2011: Snow precipitation at four ice core sites in East Antarctica: Provenance, seasonality and blocking factors. Climate Dyn., 37, 2107–2125, doi:10.1007/s00382-010-0946-4.

Figure 1 Please add insert to map showing the location of the cores.